# Coupling of remote alternating-access transport mechanisms for protons and substrates in the multidrug efflux pump AcrB

**Thomas Eicher[1,2,3,4†], Markus A Seeger[3,4,5†], Claudio Anselmi[6†], Wenchang Zhou[6,7], Lorenz Brandstätter[1,2,3,4], François Verrey[3,4], Kay Diederichs[7], José D Faraldo-Gómez[6*], Klaas M Pos[1,2*]**

[1]Institute of Biochemistry, Goethe University, Frankfurt am Main, Germany; [2]Cluster of Excellence Frankfurt, Goethe University, Frankfurt am Main, Germany; [3]Institute of Physiology, University of Zurich, Zurich, Switzerland; [4]Zurich Centre for Integrative Human Physiology, University of Zurich, Zurich, Switzerland; [5]Institute of Medical Microbiology, University of Zurich, Zurich, Switzerland; [6]Theoretical Molecular Biophysics Section, National Heart, Lung and Blood Institute, National Institutes of Health, Bethesda, United States; [7]Department of Biology, University of Konstanz, Konstanz, Germany

*For correspondence: jose.
faraldo@nih.gov (JDF-G); pos@
em.uni-frankfurt.de (KMP)

†These authors contributed equally to this work

Competing interests: The authors declare that no competing interests exist.

**Abstract** Membrane transporters of the RND superfamily confer multidrug resistance to pathogenic bacteria, and are essential for cholesterol metabolism and embryonic development in humans. We use high-resolution X-ray crystallography and computational methods to delineate the mechanism of the homotrimeric RND-type proton/drug antiporter AcrB, the active component of the major efflux system AcrAB-TolC in *Escherichia coli*, and one most complex and intriguing membrane transporters known to date. Analysis of wildtype AcrB and four functionally-inactive variants reveals an unprecedented mechanism that involves two remote alternating-access conformational cycles within each protomer, namely one for protons in the transmembrane region and another for drugs in the periplasmic domain, 50 Å apart. Each of these cycles entails two distinct types of collective motions of two structural repeats, coupled by flanking α-helices that project from the membrane. Moreover, we rationalize how the cross-talk among protomers across the trimerization interface might lead to a more kinetically efficient efflux system.

## Introduction

The AcrA-AcrB-TolC complex is a secondary active antiport system in *Escherichia coli*, powered by the electrochemical proton gradient across the inner membrane of this Gram-negative bacterium (*Okusu et al., 1996*). Its biological function is to capture cytotoxic compounds from either the periplasmic space or the inner membrane, and to expel them across the outer membrane, into the extracellular medium. This tripartite system is arguably one of the most complex and intriguing secondary transporters known to date, from both structural and mechanistic standpoints. The AcrAB-TolC system and its homologues are also of considerable biomedical interest, as they confer multidrug resistance to important pathogenic bacteria involved in human disease (*Nikaido and Pagès, 2012*).

AcrB is the inner-membrane component of the system, and is responsible for both substrate recognition and energy transduction. This membrane protein belongs to the resistance-nodulation-cell-division (RND) family (*Tseng et al., 1999*), which also includes transporters involved in protein secretion

**elife digest** The interior of living cells is separated from their external environment by an enveloping membrane that serves as a protective barrier. To regulate the chemical composition of their interior, cells are equipped with specialized proteins in their membranes that move substances in and out of cells. Membrane proteins that expel molecules from the inside to the outside of the cell are called efflux pumps.

In *Escherichia coli* bacteria, an efflux pump known as AcrB is part of a system that removes toxic substances from the bacterial cell—such as the antibiotics used to treat bacterial infections. AcrB and other closely related efflux pumps in pathogenic bacteria are often polyspecific transporters— they can transport a large number of different toxic molecules. These efflux pump systems are also more active in bacteria that have been targeted by antibiotics, and therefore they help bacteria to evolve resistance to multiple drugs. The emergence of bacterial multi-drug resistance is a global threat to human health; to combat this phenomenon, it is essential to understand its molecular basis.

Each AcrB protein has three main parts or domains. The periplasmic domain, which is located between the two membranes that surround *E. coli*, works via an 'alternating-access cycle'; that is, the shape of the periplasmic domain changes between three different forms in such a way that antibiotic molecules are first captured and subsequently squeezed through the protein towards the outside of the cell. However, the mechanism of the transmembrane domain—which is embedded in the innermost membrane of the bacterium and is the source of energy for the transport process— was not understood.

Here, Eicher et al. use X-ray crystallography to examine the three-dimensional structures of the AcrB efflux pump—and several inactive variants—in high detail. Combining these results with computer simulations reveals the mechanism used by the transmembrane domain to take up protons from the exterior and transport them into the cell. Proton transport also proceeds according to an alternating-access mechanism—and, although the transmembrane and periplasmic domains are far apart, their movements are tightly linked. Thus, because proton uptake releases energy, the transmembrane domain effectively powers the periplasmic domain to expel drugs and other molecules. Eicher et al. note that a similar mechanism has not been seen before in other efflux pumps or transporter proteins.

Understanding how AcrB works opens up new avenues that could be exploited to develop new drugs against multidrug resistant bacteria. Furthermore, Eicher et al. suggest that efflux pumps in humans closely related to AcrB might function in a similar way—including those required for regulation of cellular cholesterol, and for the correct development of embryos.

(e.g., SecDF of *E. coli*), export of lipids (e.g., MmpL7 of *Mycobacterium tuberculosis*) and pigment export (e.g., ORF4 of *Xanthomonas oryzae*). In humans, the RND transporters NPC1 and Patched play key roles in the regulation of cholesterol metabolism and in the Hedgehog signaling pathway of embryonic cells, respectively (*Taipale et al., 2002*; *Scott and Ioannou, 2004*).

The first known structure of AcrB, solved by X-ray crystallography at a resolution of 3.5 Å, revealed a homotrimeric assembly in which all protomers adopt the same conformation (*Murakami et al., 2002*). In each protomer, the transmembrane (TM) domain consists of 12 α-helices (TM1–TM12), mostly interconnected by short unstructured loops as is common. In contrast to other secondary transporters, however, AcrB also features two large periplasmic insertions, between TM1 and TM2, and between TM7 and TM8, which account for approximately 60% of the 1049 amino acids in each protomer. These insertions are highly structured domains that project 70 Å away from the membrane, and form extensive interfaces among protomers (*Figure 1*). The TM domains barely contribute to the trimer interface, in further contrast to typical homomeric transporters. The more distal region of the periplasmic domain features a central funnel-shaped cavity that channels the substrate towards TolC, the outer-membrane channel component (140 Å in length) of this transport system (*Koronakis et al., 2000*). AcrA, the third component, functions as a peripheral scaffold to mediate the interaction between AcrB and TolC (*Akama et al., 2004*; *Mikolosko et al., 2006*; *Tikhonova et al., 2011*; *Du et al., 2014*).

The intricate and unusual topology of AcrB and its trimerization state were confirmed by subsequent higher-resolution X-ray structures obtained from crystals grown in different space groups,

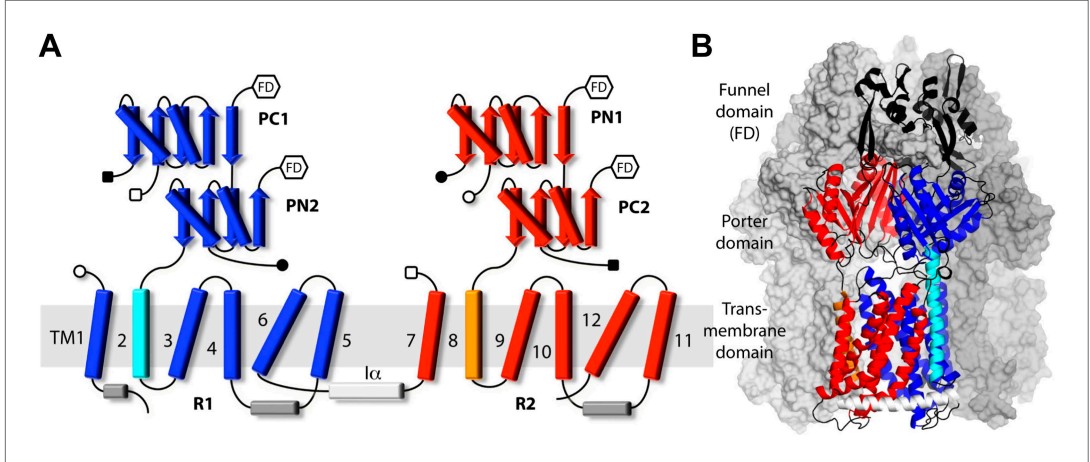

**Figure 1**. Topology and secondary structure representations of the AcrB protomers. (**A**) Topology of the AcrB protomers in the transmembrane (TM) and porter domains. Our analysis indicates that the TM domain consists of two 5-helix parallel repeats, referred to as R1 and R2, and two flanking helices, TM2 and TM8. R1 (blue) is connected to R2 (red) through helix Iα (white), which lies parallel to the cytoplasmic face of the membrane. The porter domain also consists of two repeats of two subdomains each, namely PC1 and PN2 (blue), and PN1 and PC2 (red). TM2 (cyan) connects R1 to PN2, whereas TM8 (orange) links R2 to PC2. Other connections among porter sub-domains and R1/R2 are indicated by open/closed squares/circles. Hexagons indicate the connections between the porter domain and the funnel domain (FD), not drawn in this scheme. (**B**) Cartoon representation of the secondary structure of an AcrB protomer, in the context of the complete trimer. The other two protomers are drawn as a molecular surface (gray). Different sub-domains are colored as in (**A**), with inter-connecting loops and the FD shown in black.

The following figure supplements are available for figure 1:

**Figure supplement 1**. Cartoon representation of the porter domain of AcrB in the asymmetric state, viewed from the outer membrane, along the perpendicular to the inner membrane.

**Figure supplement 2**. Molecular simulation systems employed in this study.

---

several of which include bound substrates (*Murakami et al., 2006*; *Seeger et al., 2006*; *Sennhauser et al., 2007*; *Nakashima et al., 2011*, *2013*; *Eicher et al., 2012*). These structures, however, also revealed that upon substrate loading AcrB becomes asymmetric, that is, each protomer in the trimer adopts one of three distinct conformational states, designated as 'loose' (L), 'tight' (T) and 'open' (O), or alternatively, as 'access', 'binding' and 'extrusion', (*Murakami et al., 2006*; *Seeger et al., 2006*). On the basis of these structures as well as other findings (*Seeger et al., 2008*; *Takatsuka and Nikaido, 2009*), it has also become generally accepted that drug recognition and efflux are mediated by a subdomain of the periplasmic region, referred to as the porter domain (*Figure 1*, *Figure 1—figure supplement 1*), whose mechanism is akin to that of a peristaltic pump (*Seeger et al., 2006*). Specifically, recognition of high molecular-mass drugs occurs in the L state, via an access pocket in the porter domain that is open to the periplasmic space (*Nakashima et al., 2011*; *Eicher et al., 2012*). When the L protomer transitions to the T state, drugs translocate to a binding pocket deeper in the protein; low molecular-mass drugs, however, can apparently bind directly to the deep binding pocket in the T protomer without binding to the access pocket first (*Nakashima et al., 2011*). Drug extrusion occurs next, in the transition from T to O, which re-shapes and opens the binding pocket to a central cavity in the funnel domain, which leads to the lumen of the TolC channel. Finally, the O to L transition resets the conformational cycle by re-exposing the binding site to the periplasm. The extensive interface between protomers in the porter domain, and the asymmetric nature of the trimer, also suggest that the three protomers are conformationally coupled (*Pos, 2009*). Indeed, recent measurements of AcrB transport kinetics (*Nagano and Nikaido, 2009*; *Lim and Nikaido, 2010*) are consistent with a bi-site activation model (*Pos, 2009*), analogous to the binding-change mechanism of the ATP synthase (*Boyer, 1997*).

The transport mechanism of the TM domain of AcrB, in contrast, remains largely undefined. The asymmetric trimer structures indicate that this domain also cycles between three distinct conformations, and point to several ionizable residues in its core (D407, D408, K940 and R971) that might function as a proton-relay network. Nevertheless, no clear interpretation of existing data has been formulated yet in terms of an alternating-access proton transport mechanism, which is a necessary condition for the transporter to harness the energy stored in the transmembrane electrochemical gradient. How this proton-motive-force is mechanically transduced to the porter domain, in order to power uphill drug efflux, also remains to be determined.

Here, we gain insights into these crucial mechanistic questions from a series of new crystal structures of AcrB, systematically analyzed with molecular simulations and other computational methods. More specifically, we examine the TM domain of the transporter in an improved atomic structure of wildtype AcrB (*Eicher et al., 2012*), at 1.9 Å resolution, and also report the structures of four single-substitution variants (D407N, D408N, K940A and R971A), all of which are inactive both in proton translocation and drug efflux, solved at resolutions between 2.0 and 2.3 Å. Taken together with our computational analysis, these data enable us to delineate the alternating-access mechanism of proton transport of the TM domain, and rationalize how this mechanism energizes drug efflux, more than 50 Å away.

## Results and discussion

### Modularity and collective motions in the periplasmic and transmembrane domains

Despite the unique topological complexity of AcrB, a systematic analysis of its structure permits to rationalize its conformational cycle in terms of collective motions of distinct structural units. In the porter domain, as noted previously, each protomer consists of two structural repeats, referred to as PN1/PC2, and PN2/PC1 (*Figure 1*, *Figure 1—figure supplement 1*). Each repeat is a tandem of two similar α/β sub-domains, joined by a shared β-strand. The PN1/PC2 repeat appears to be a rigid unit throughout the LTO conformational cycle. Analysis of the conformation of this repeat in the 1.9 Å-resolution crystal structure of wildtype AcrB (*Eicher et al., 2012*) yields root-mean-square differences (RMSD) among states of less than 0.5 Å (*Supplementary file 1*). By contrast, the PN2/PC1 tandem changes its conformation significantly, in particular in the L to T (RMSD ~2.0 Å) and T to O transitions (RMSD ~1.8 Å). This is due to the rearrangement of PN2 relative to PC1, rather than to changes in the internal structure of the individual α/β domains, which appear to be quite inflexible across the cycle, in both repeats (*Supplementary file 1*). As described elsewhere (*Murakami et al., 2006*; *Seeger et al., 2006*; *Sennhauser et al., 2007*; *Pos, 2009*; *Nakashima et al., 2013*), the changes in PN2/PC1 reflect binding of the substrate deep within the hydrophobic core of the tandem, in the T state, and its subsequent release into the interior of the funnel domain, in the O state (*Figure 1—figure supplement 1*; *Video 1*, *Video 2*, *Video 3*). Incidentally, the conformation of the funnel domain is largely unchanged throughout the cycle (RMSD < 0.5 Å).

Given the modular nature of the porter domain, we reasoned that the TM domain might also consist of structural repeats, as is observed in other secondary transporters. To identify these, we quantified the degree of similarity of combinative structural superpositions of fragments of the transmembrane 12-helix bundle in the L, T and O states (*Figure 2*). This analysis indicates that the conformational cycle of the TM domain entails relative motions of two structural repeats within each protomer, which had not been previously recognized. Each of these repeats, which we refer to as R1 and R2, comprise five transmembrane α−helices. The N-terminal repeat, R1, includes TM1 and TM3 to TM6, while the C-terminal repeat, R2, comprises TM7 and TM9 to TM12 (*Figure 1*). R1 and R2 are thus oriented in parallel, and are rotated relative to each other approximately 180° around an axis perpendicular to the membrane plane. Interestingly, each repeat is flanked by a single transmembrane helix that seems to function as a coupling element with the periplasmic porter domain (*Figure 1*). The flanking helices are TM2 and TM8, respectively; TM2 is linked to the flexible PN2/PC1 repeat in the porter domain, while TM8 is connected to the more rigid PN1/PC2 unit (*Figure 1*).

Analysis of R1, for example, shows that the structure of this five-helix bundle is largely preserved in the L, T and O conformations; the RMSD among these states ranges from 0.8 to 1.1 Å (*Figure 2—figure supplement 1*). These differences are even smaller for R2, with RMSD values between 0.3 and 1.0 Å (*Figure 2—figure supplement 1*). R1 and R2 are, however, always different from each other

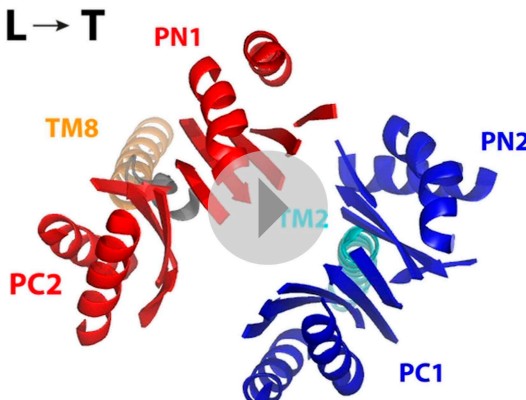

**Video 1**. Structural repeats within the periplasmic porter domain of AcrB, PN1/PC2 and PN2/PC1, and their collective motions during the transport cycle. L to T transition.

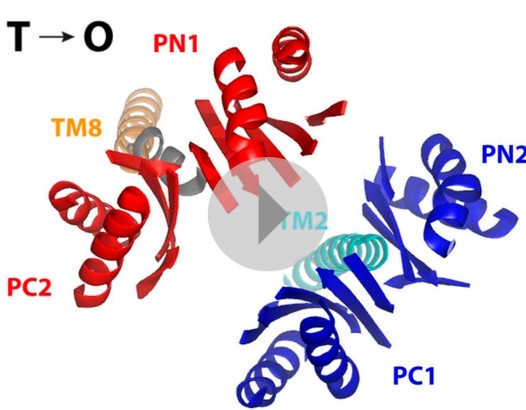

**Video 2**. Structural repeats within the periplasmic porter domain of AcrB, PN1/PC2 and PN2/PC1, and their collective motions during the transport cycle. T to O transition.

(RMSD ~3.5 Å). Further inspection also shows that R1 and R2 move relative to each other throughout the conformational cycle of each protomer (*Figure 2—figure supplement 2*). For example, if we superimpose R1 in the L and T conformations, the RMSD of the corresponding R2 repeats is 1.6 Å; for the T to O transition the analogous value is 2.1 Å, and for O to L, 3.1 Å. These relative differences are substantially larger than those observed individually within R1 or R2, for all steps of the cycle. Therefore, it is clear that R2 and R1 undergo collective motions during the LTO cycle that are larger in magnitude than the changes in their internal structure.

Structural overlays and interpolations of the L, T and O conformations, based on the topological decomposition just described, reveal the nature of the relative repeat motions within the TM domain during the transport cycle (*Figure 3*; *Video 4*, *Video 5*, *Video 6*). In the L to T transition, R1 and R2 undergo a rocking motion around an axis that is tilted relative to the membrane plane (*Figure 3*; *Video 4*). By contrast, in the transition between the T and O states, and between O and L, what can be observed is mostly a lateral shear motion of R1 and R2, relative to each other (*Figure 3*; *Video 5*, *Video 6*). This analysis also highlights the motions in the helices flanking the repeats, TM2 and TM8. In the L to T transition, TM2 moves downwards relative to R1 (toward the cytoplasm), returning to the up state in the O conformation (*Figure 3*; *Video 4*, *Video 5*, *Video 6*). By contrast, TM8 is similarly positioned in the L and T states, and largely unengaged from R2. However, during the T to O transition this α-helix becomes closely packed against R2, through hydrophobic interactions with TM5. In this transition TM8 also becomes elongated, reaching directly to the PN1/PC2 unit (*Figure 3*; *Video 4*, *Video 5*, *Video 6*).

Interestingly, the membrane domain features an additional helix, Iα, which lies parallel to the membrane, on the cytoplasmic side. This helix is inserted between TM6 and TM7 in the protein sequence, and thus links R1 and R2 (*Figure 1*). Structurally, however, Iα is mostly adjacent to R2, and interacts with R1 (via hydrophobic interactions and hydrogen-bonds) primarily at the cytoplasmic end of TM2, at a right angle (*Figure 3*; *Video 4*, *Video 5*, *Video 6*). Our analysis shows that Iα is effectively part of R2, and that it moves rigidly along with this repeat in the three conformational transitions, most probably in response to the motion of TM2 (*Figure 3—figure supplement 1*). This notion is consistent with the finding that MexB, a homolog of AcrB, is functional if the *mexB* gene is expressed in two halves, split at the N-terminal end of Iα (*Eda et al., 2003*). This construct lacks a direct link between Iα and TM6 in R1, but likely preserves the large interface between Iα and R2, as well as the interaction with TM2.

## The membrane-domain repeats switch between an engaged and a disengaged state

Previous crystal structures of AcrB (*Murakami et al., 2006*; *Seeger et al., 2006*; *Sennhauser et al., 2007*; *Eicher et al., 2012*) have revealed an intriguing network of ionic and polar side-chains in the core

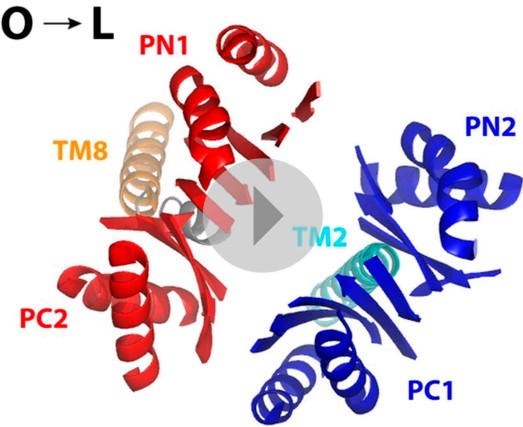

**Video 3**. Structural repeats within the periplasmic porter domain of AcrB, PN1/PC2 and PN2/PC1, and their collective motions during the transport cycle. O to L transition.

of the TM domain, which seems to re-configure as the TM domain cycles between conformations. In particular D407 and D408 in TM4 and K940 in TM10 are critical for function (*Guan and Nakae, 2001*; *Murakami and Yamaguchi, 2003*; *Takatsuka and Nikaido, 2006*; *Seeger et al., 2009*). Another functionally important side-chain, R971 in TM11, is located about 10 Å away from K940 on TM10, towards the cytoplasm.

Interestingly, all these side chains are found at the interface between R1 and R2, seemingly controlling the relative position of the repeats. The latest high-resolution (1.9 Å) crystal structure of AcrB (PDB entry 4DX5, *Eicher et al., 2012*), whose TM domain had not been previously analyzed, reveals in detail how this side-chain network reorganizes throughout the conformational cycle (*Figure 4*). In both the L and T states, R1 and R2 appear to be strongly paired through multiple interactions. First, through K940, which projects away from R2 and is sandwiched between D407 and D408, from R1, forming two simultaneous ionic interactions. D407 is the acceptor of an additional hydrogen-bond between repeats, namely from T978 in TM11. And lastly, towards the cytoplasm, R971 is in close proximity to E414 and N415, from TM4, particularly in the T state (*Figure 4*). (D408 is also the acceptor of an additional hydrogen-bond, but within its own repeat, namely S481 from TM6.) In the O state, by contrast, all R1-R2 interactions break off, and are replaced by alternative ones within each repeat. D407 rotates to engage the backbone of G403, one helix-turn above along TM4; K940 retracts towards R1, and donates hydrogen-bonds to N941 (TM10) and T978; and R971 moves away from E414/N415, and interacts instead with the backbone carbonyl of L944 and the aromatic ring of F948, both in TM10. (D408 also switches interaction partners, to the backbone of L442 in TM5, i.e. also within repeat R1.) In sum, repeats R1 and R2 appear to exchange between an 'engaged' state in the L and T conformations, to a 'disengaged' state, in the O conformation.

## Proton relay across the membrane domain

To examine how the local and global changes in the structure of the TM domain of AcrB are related to proton translocation, we carried out two distinct but complementary computational analyses. First, we conducted a series of Monte Carlo simulations of the protonation equilibrium of all aspartate, glutamate and histidine side-chains in the protein under a continuum-electrostatics framework, that is, via the Poisson equation ('Materials and methods'). In these calculations all lysine and arginine side-chains were assumed to be ionized, except for K940 and R971, which were explicitly analyzed. *Figure 5* summarizes the results from these simulations, focusing on the TM domain. Most of the aspartate, glutamate and histidine side-chains were found to be deprotonated across the LTO cycle; many of these aspartate and glutamate are engaged in salt-bridges, particularly on the cytoplasmic side of the protein (*Figure 5A,B*). A handful of residues, however, change their most probable protonation state in the T to O transition. These include D407 and D408, in the core of the TM domain, which are ionized in the L and T states but become protonated in the O state (*Figure 5C*). Interestingly, E346 and D924, at the periplasmic end of TM2 and TM10, respectively, are protonated in the L and T states, but become deprotonated in O; conversely, the neighboring H338 (TM2) and D566 (in the region between TM7 and PC1 subdomain) are most likely deprotonated in L and T, and become protonated in the O state (*Figure 5C*). Lastly, the calculations indicate that both K940 and R971 remain protonated throughout the cycle.

As is well known, electrostatic-energy calculations based on the Poisson equation often yield divergent results depending on the value of the protein dielectric constant that is assumed. Although the results just described are generally consistent across the range of most plausible values (*Kukic et al., 2013*), namely 2 to 6, we sought to confirm our conclusions with a second methodology. In particular we used a series of all-atom molecular dynamics simulations of AcrB protomers embedded in a

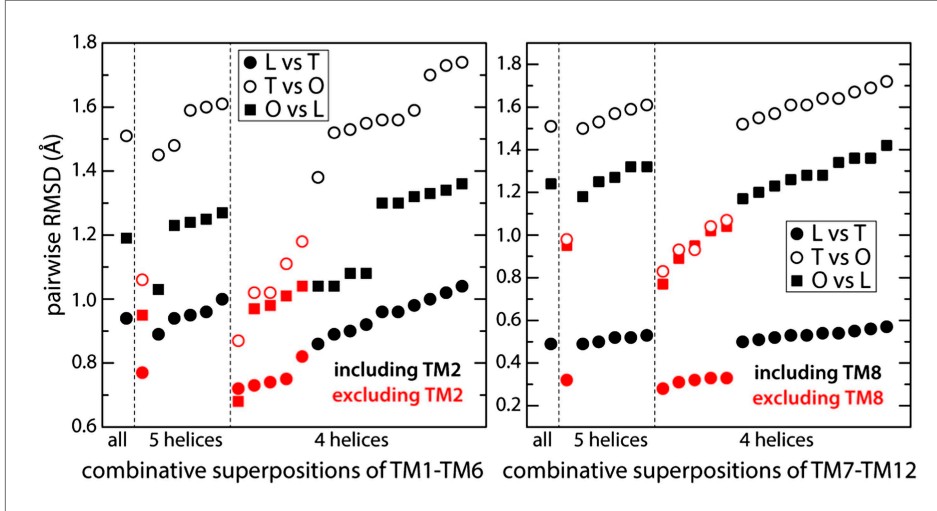

**Figure 2**. Identification of structural units within the TM domain of AcrB undergoing collective motions throughout the LTO conformational cycle. We considered all possible combinations of 6, 5 and 4 transmembrane helices, within either TM1-TM6 or TM7-TM12, and compared the structure of each set of helices in the L vs T states (closed circles), T vs O (open circles) and O vs L (closed squares), through pairwise root-mean-square difference (RMSD) calculations. When these RMSD values are used as a rank, it becomes clear that combinations that exclude TM2 or TM8 (red) feature systematically smaller differences across the LTO cycle than those that include TM2 or TM8 (black). All other exclusions show no clear correlation. For example, comparing TM1-TM6 (left panel) in L vs T, this increasing ranking is ΔTM2, ΔTM4, ΔTM1, ΔTM5, ΔTM3, and ΔTM6; in T vs O: ΔTM2, ΔTM5, ΔTM4, ΔTM3, ΔTM1, and ΔTM6; and in O vs L: ΔTM2, ΔTM5, ΔTM1, ΔTM4, ΔTM3, and ΔTM6. Furthermore, the data shows that combinations of five helices excluding TM2 and TM8 are not significantly more divergent across the LTO conformers than combinations of four helices. Thus, the asymmetric crystal structure of AcrB captures collective displacements of two 5-helix bundles within each protomer, formed by TM1/TM3-TM6 and TM7/TM9-TM12 (referred to as repeats R1 and R2), relative to flanking helices TM2 and TM8, respectively. Note that comparisons of combinations of seven or more helices result in larger RMSD values than those shown here, and are therefore omitted.

The following figure supplements are available for figure 2:

**Figure supplement 1**. Structural variations in the R1 and R2 repeats of the transmembrane domain of wildtype AcrB, in the L, T and O states.

**Figure supplement 2**. Relative displacements of structural elements in the transmembrane and porter domain of wildtype AcrB across the LTO conformational cycle.

---

phospholipid membrane (*Figure 1—figure supplement 2*) to calculate the protonation free energy of individual sites, relative to side-chain analogs in bulk water ('Materials and methods'). Specifically, we computed the free energy of protonation of D407, D408 and K940, in the O state, and of E346 and D924, in the T conformation. The results, shown in *Figure 6*, reveal strong upward shifts in the p$K_a$ of D407 (in the O conformation), D408(O), E346(T) and D924(T), and a weak downward shift in K940(O), thus confirming that these side-chains are most likely protonated in those protomer states. As controls we also analyzed E693(T) and K603(O), which are located on the solvent-exposed surface of the periplasmic PC2 subdomain; as expected, the calculated p$K_a$ shifts for these side-chains are small or negligible (*Figure 6*). In sum, both all-atom molecular simulations and continuum-electrostatic calculations indicate that in the L and T states, two protons are bound to the TM domain of AcrB at the periplasmic side (i.e., D924 and E346), while D407 and D408, two-thirds of the way across the membrane, are deprotonated; upon transition to the O state, two protons bind to D407 and D408, while the proton-accepting sites on the periplasmic face become deprotonated.

The proposed changes in the protonation state of D407 and D408 in the transitions from T to O (protonation) and O to L (deprotonation) are consistent with the 'disengagement' and 'reengagement', respectively, of the structural repeats R1 and R2, as the ionic interactions with K940 would be weakened by protonation of the aspartate side-chains. The finding that K940 remains ionized in the

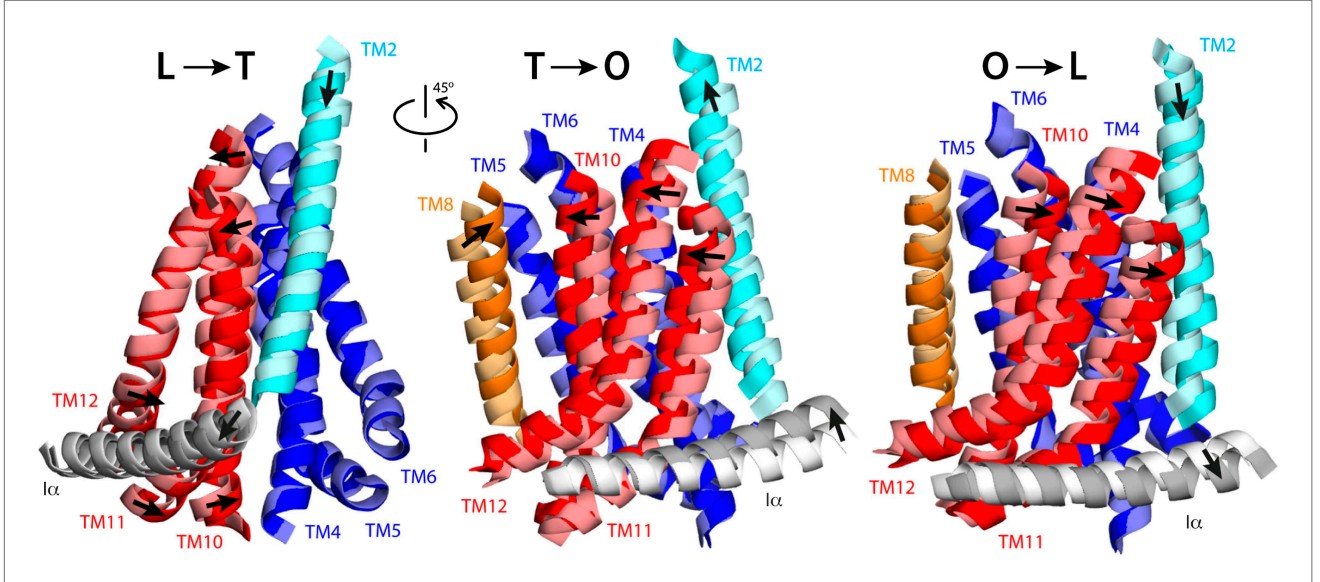

**Figure 3**. Collective motions of the structural repeats in TM domain of AcrB. For clarity, the figure shows only the transmembrane α-helices at the interface between repeats R1 (blue) and R2 (red), alongside the two α-helices flanking these repeats, TM2 (cyan), TM8 (orange), and the helix that links the two repeats, Iα (gray). Each panel depicts two conformations in the LTO cycle, overlaid optimally on TM4-TM6; thus the figure highlights the motions of R2 relative to R1. Initial and final states (e.g., L vs T) are shown in light and bright colors, respectively. Note that the L vs T overlay is viewed from a different angle than T vs O and O vs L, for clarity. Interpolations between these conformational states using the complete TM domain are shown in *Video 4*, *Video 5* and *Video 6*.

The following figure supplement is available for figure 3:

**Figure supplement 1**. Arrangement of helix Iα across the conformational cycle of wildtype AcrB.

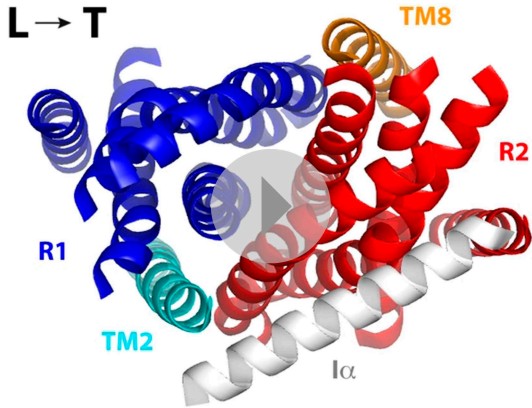

**Video 4**. Structural repeats within the transmembrane domain of AcrB, R1 and R2, and their collective motions during the transport cycle. L to T transition.

O state might be less intuitive. However, analysis of non-redundant high-resolution structures in the Protein Data Bank reveals thousands of instances in which a lysine side-chain is part of an electrostatic interaction network that does not include any negatively-charged contacts (*Figure 6—figure supplement 1*). Moreover, in this O state K940 not only hydrogen-bonds with N941 with T978 (*Figure 4*), but also interacts with water, as we will show below. This polar micro-environment explains the small shift in the p$K_a$ calculated here. Consistent with this result, measured p$K_a$ values for lysine side-chains engineered in a slightly less polar (and also non-ionic) environment within the model protein SNase are down-shifted only by ~2 units (*Damjanović et al., 2011*).

As mentioned above, experimental evidence demonstrates that K940 and R971 are indispensable for the proton-driven mechanism of AcrB (*Guan and Nakae, 2001*; *Murakami and Yamaguchi, 2003*; *Takatsuka and Nikaido, 2006*; *Seeger et al., 2009*). Our results suggest that the primary role of these side-chains is not, however, to capture and donate protons; instead, they appear to function as a conformational electrostatic switch coupled to the protonation state of D407 and D408. In the L and T states both K940 and R971 have strong favorable interactions with ionized D407 and D408, which exceed their mutual repulsion (*Figure 5—figure supplement 1*). Neutralization of D407 and D408, in the transition to the O state, alters this

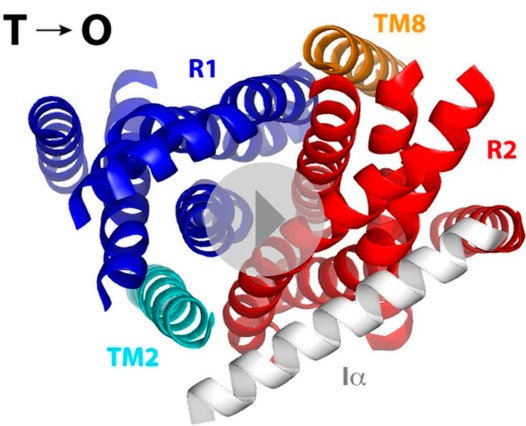

**Video 5**. Structural repeats within the transmembrane domain of AcrB, R1 and R2, and their collective motions during the transport cycle. T to O transition.

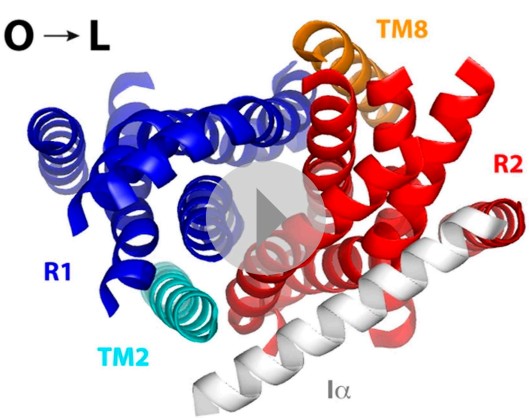

**Video 6**. Structural repeats within the transmembrane domain of AcrB, R1 and R2, and their collective motions during the transport cycle. O to L transition.

electrostatic balance. As a result, K940 and R971 reorient to reduce their electrostatic repulsion and to maximize alternative favorable interactions within the protein, particularly within repeat R2 (*Figure 5*, *Figure 5—figure supplement 1*). We propose that the re-configuration of this reversible electrostatic switch, which is coupled to the protonation and deprotonation of both D407 and D408, is the trigger for the lateral displacements of repeat R2 relative to R1 observed in the T to O and O to L transitions (*Figure 3*; *Video 5*, *Video 6*).

## Structures of AcrB variants are consistent with a two-proton conformational-switch mechanism

To further assess the proposed proton-relay mechanism and its influence on the conformation of the TM domain, we solved the atomic structures of four AcrB variants, namely K940A, R971A, D407N and D408N (*Supplementary file 2*), all of which abolish the drug-efflux activity of AcrB (*Guan and Nakae, 2001*; *Murakami and Yamaguchi, 2003*; *Takatsuka and Nikaido, 2006*; *Seeger et al., 2009*). Electron-density maps for these variants could be resolved at resolutions ranging from 2.0 to 2.3 Å, which are sufficient to discern the precise configuration of side-chains and structural waters inside the TM domain.

In all cases, the crystal structures represent an asymmetric configuration of the AcrB trimer. As in the wildtype transporter, this asymmetry originates in the porter domain due to drug binding (*Supplementary file 3A*, *Supplementary file 3B*), compounded by the steric coupling between protomers in this region. This asymmetry in the porter domain is important for crystallization, as it facilitates the recognition of designed-ankyrin repeats proteins (DARPins) developed to improve wildtype crystals (*Sennhauser et al., 2007*). As in wildtype AcrB, the asymmetry of the porter domain propagates to the transmembrane region, where the protomers also adopt three distinct conformations.

None of the substitutions studied caused significant changes in the internal structure of the R1 and R2 repeats, compared to the wildtype L, T and O states (*Supplementary file 3A*). The relative orientation of these two repeats is also unchanged in the K940A and D408N mutants (*Supplementary file 3C*), as is the side-chain configuration of the proton-relay network, aside from the mutations (*Figure 7*). That is, these two substitutions appear to perturb the conformational cycle of the transporter by altering the energetic balance between the L, T and O states, rather than by precluding the protomers from adopting native-like conformations in each of these states. Indeed, electrostatic calculations analogous to those carried out for the wildtype protein show that in the K940A structure, D408 is constitutively neutralized throughout the cycle. D407, which is more proximal to R971, is deprotonated in the L and T states and protonated in the O state, as in the wildtype (*Figure 7*, *Figure 7—figure supplement 1*). Similarly for the D408N variant, D407 is predicted to become protonated only in the O state, while the substituted N408 is evidently neutral throughout the cycle (K940 and R971 are protonated in all states in both mutants) (*Figure 7*). These results imply that the ionic interactions across the R1-R2 interface that stabilize the L and T states are weakened in the mutants. By contrast, the O state is probably less affected (K940A) or not at all (D408N), since the repeats

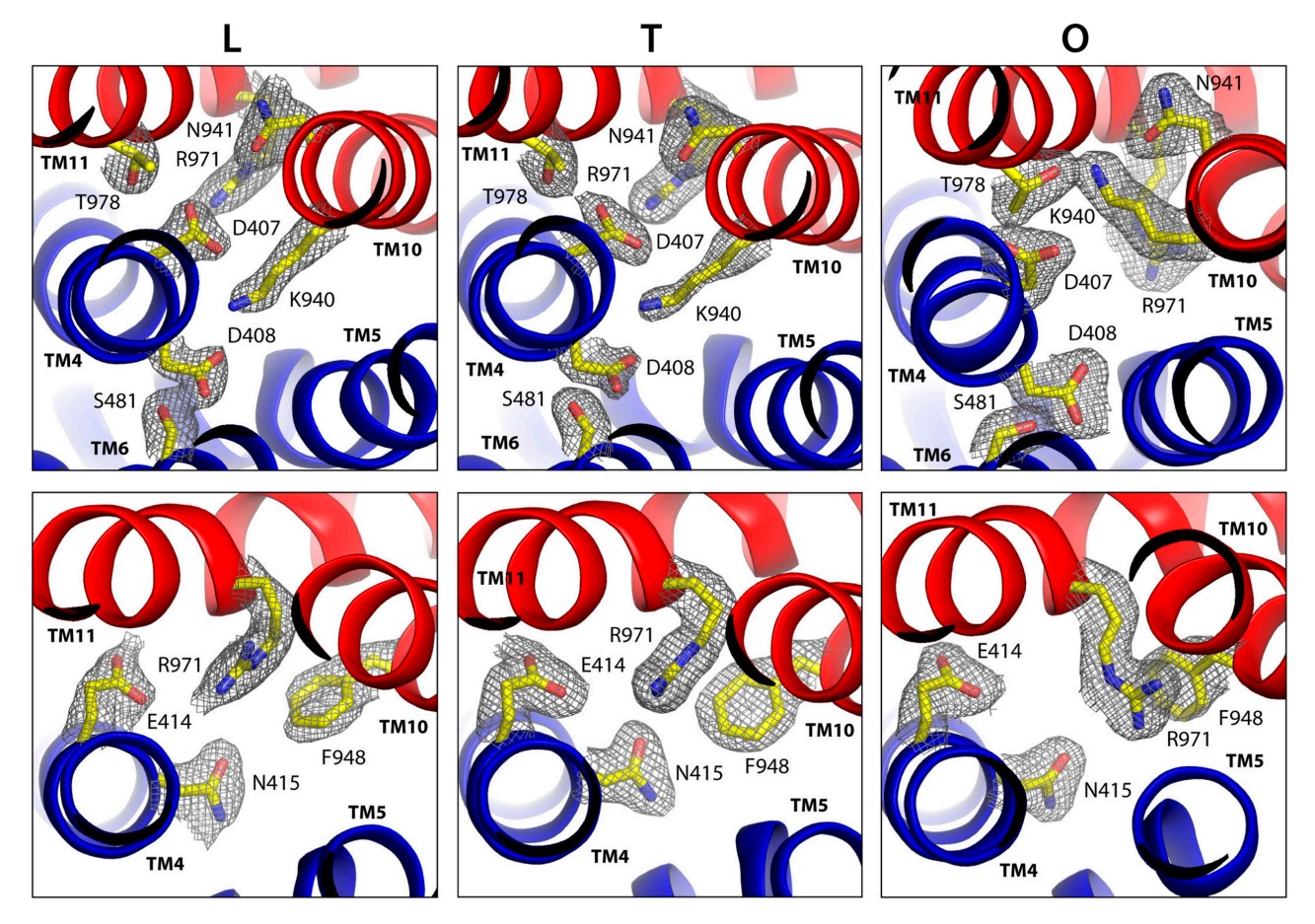

**Figure 4**. Configuration of the proton-dependent conformational switch in the core of the TM domain of AcrB, in the L, T and O states, viewed from the periplasm. The structure corresponds to the asymmetric crystal structure of AcrB at a resolution of 1.9 Å (PDB entry 4DX5). The top panels show a close-up of the proton-binding sites, D407 and D408, along with neighboring side-chains, including K940 in the foreground and R971 in the background. The lower panels are close-ups of R971 and interacting side-chains in its proximity. The highlighted side-chains are shown along with 2F$_o$-F$_c$ electron density maps contoured at 1.0σ. Note this proton-relay network is precisely at the interface between the two transmembrane repeats, R1 and R2 (blue and red). DOI: 10.7554/eLife.03145.017

become disengaged in this state. This change in the energetic balance of the conformational cycle (which for example would hinder the transition from O to L) would explain the drastic functional effect of these two mutations.

Electrostatic calculations for the R971A and D407N crystal structures indicate that, like K940A and D408N, these variants lack the ability to bind and release two protons (*Figure 7—figure supplement 1*). In R971A, like in K940A, a first proton is already bound to the protomer in the L and T states (alternating between D407 and D408, respectively) and a second proton binds in the O state (*Figure 7*). In D407N, like in D408N, only one proton is loaded (on D408), and only in the O state (*Figure 7*). Thus, also in these two variants is the key ionic interaction network at the R1-R2 interface partially neutralized in the L and T states. The R971A and D407N substitutions, however, not only perturb the energetics of the conformational cycle and the transduction of the proton electrochemical gradient, but also seem to preclude the protomers from adopting all the native-like conformations, particularly the T state (*Figure 7*). The structure of R971A shows that K940 is engaged only with D408 in the L state (while D407 is protonated). In the T state, K940 switches towards D407 (while D408 is protonated), adopting a configuration that is arguably an intermediate between the native T and O states: first, in that K940 retracts away from the repeat interface, unhindered by the now absent repulsion with R971, while still paired with D407; and second, in that as a result D407 loses its hydrogen bond to T978 across the R1-R2 interface. Consistent with our hypothesis that the structural

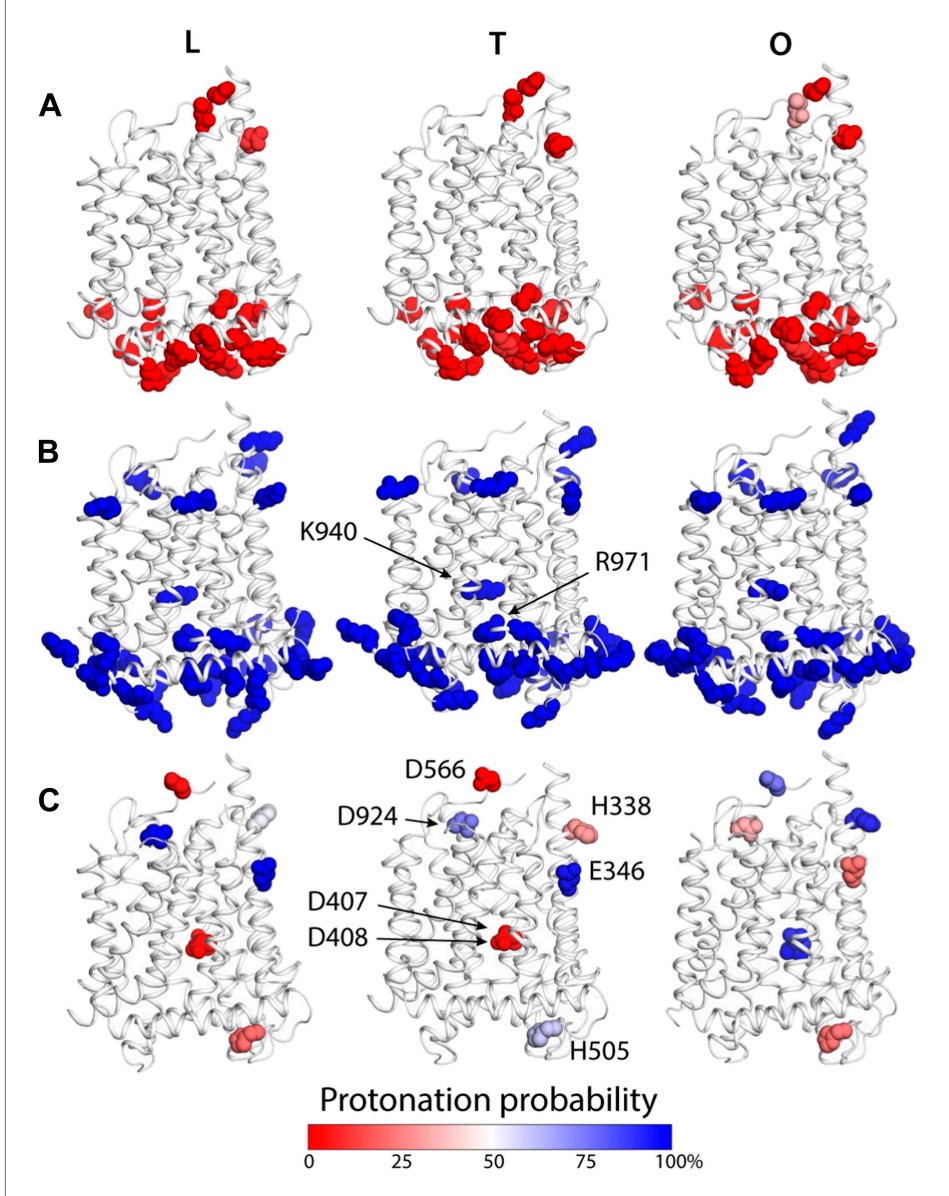

**Figure 5**. Changes in the computed protonation probability of ionizable residues in the TM domain, across the LTO conformational cycle. The three protomers in the asymmetric crystal structure of AcrB are shown side by side, in a cartoon representation (white), in the same orientation as in *Figure 1*. Ionizable residues are represented as van-der-Waals spheres, and colored according to their protonation probability, computed via Monte-Carlo Poisson–Boltzmann calculations ('Materials and methods'). (**A**) Glu, Asp and His residues predicted to be in the deprotonated state in all conformations of the protomer. (**B**) K940 and R971 are predicted to be protonated in the L, T and O states; other lysine and arginine residues are assumed to be protonated in the calculation. (**C**) Glutamate, aspartate and histidine residues whose protonation state is predicted to be inter-dependent with the conformation of the protomer.

The following figure supplement is available for figure 5:

**Figure supplement 1**. Electrostatic switch involving K940 and R971 upon protonation of D407 and D408.

configuration of the proton-relay network is intimately coupled to the displacements of the 5-helix repeats, the arrangement of these repeats in the T protomer of the R971A crystal structure is an intermediate between the T and O states of the wildtype structure (*Figure 8*); in fact, this altered conformation is more similar to the latter (RMSD values of 2.4 vs 1.6 Å, respectively—*Supplementary file 3C*).

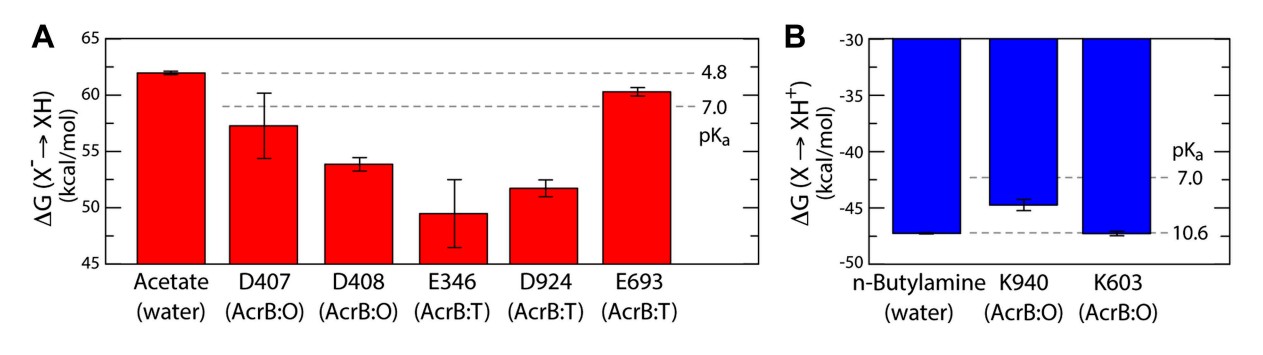

**Figure 6**. Free energy of protonation for selected glutamate, aspartate and lysine residues in the L, T and O states. All values were computed via all-atom molecular simulations in a phospholipid membrane ('Materials and methods'). Computed values for side-chain analogs in solution, whose p$K_a$ is known experimentally, provide a means to translate the computed free-energy values into a p$K_a$ scale (dashed lines). (**A**) The calculations indicate upward shifts in the p$K_a$ of E346 and D924 in the T state, and of D407 and D408 in the O state, relative to acetate in water, whose magnitude indicate that in those conformational states these residues are protonated at physiological pH. By contrast, the p$K_a$ of E693 is hardly shifted, consistent with its location on a water-exposed, electrostatically neutral area of the protein surface. (**B**) The p$K_a$ of K940 in the O state is also shifted downwards relative to n-butylamine in solution, on account of its being largely unexposed, but does not become deprotonated owing to polar contacts with N941 and T978 (**Figure 4**).

The following figure supplement is available for figure 6:

**Figure supplement 1**. Statistical analysis of ionic and non-ionic side-chains interactions formed by lysine side-chains in high-resolution crystal structures deposited in the Protein Data Bank.

The effect of the D407N substitution is subtler than that of R971A, but also underscores the relationship between the local configuration of the proton-relay network and the organization of the transmembrane repeats. The D407N structure shows that although the substituted asparagine can accept a hydrogen-bond from K940 in the L and T states, its orientation is geometrically inconsistent with the second hydrogen-bond with T978 seen in the wildtype (**Figure 7**). In the L state this appears to be compensated by an alternative hydrogen-bond donated by N941, but this interaction is not feasible in the T state. In addition, the asparagine substitution removes the attractive electrostatic interaction between D407 with R971. In sum, the interaction network at the R1-R2 interface is partially disengaged. Consequently, the T state in the D407N structure shows R1 and R2 partially displaced, in a conformation that is again intermediate between the wildtype T and O states (**Figure 8**; **Supplementary file 3C**).

As mentioned above, the crystallization of these variant structures was facilitated by DARPin ligands, which recognize the porter domain of AcrB in its asymmetric form (**Sennhauser et al., 2007**; **Eicher et al., 2012**). Because the porter and transmembrane domains are necessarily coupled, it is conceivable that the DARPins inhibit the structural impact of the mutations, that is, only the strongest perturbations (R971A and D407N) would translate into noticeable structural differences in our experiment. Thus, we cannot rule out that the lack of activity of these mutants reflects structural perturbations more significant than those observed here. Nonetheless, even the relatively modest changes revealed by the structures presented here support the conclusions of our analysis of wildtype AcrB. That is, that there is a causal relationship between the displacements of the repeats R1 and R2 within the TM domain and the configuration of the proton-dependent electrostatic network at the R1-R2 interface; that there is a conformational and electrostatic coupling between R971 and K940 and the two aspartate side-chains that serve as proton-binding sites; and lastly, that the transitions from T to O and O to L, which power the peristaltic pump mechanism of the porter domain, entail binding and release of two protons at D407 and D408.

## Hydration of the membrane domain reveals alternating access to proton-relay site

Proton translocation across membrane proteins necessarily requires a pathway of water molecules to facilitate access from the surrounding solution, and to mediate proton hopping between protonatable side-chains. In the case of proton-driven secondary transporters, these pathways must also be

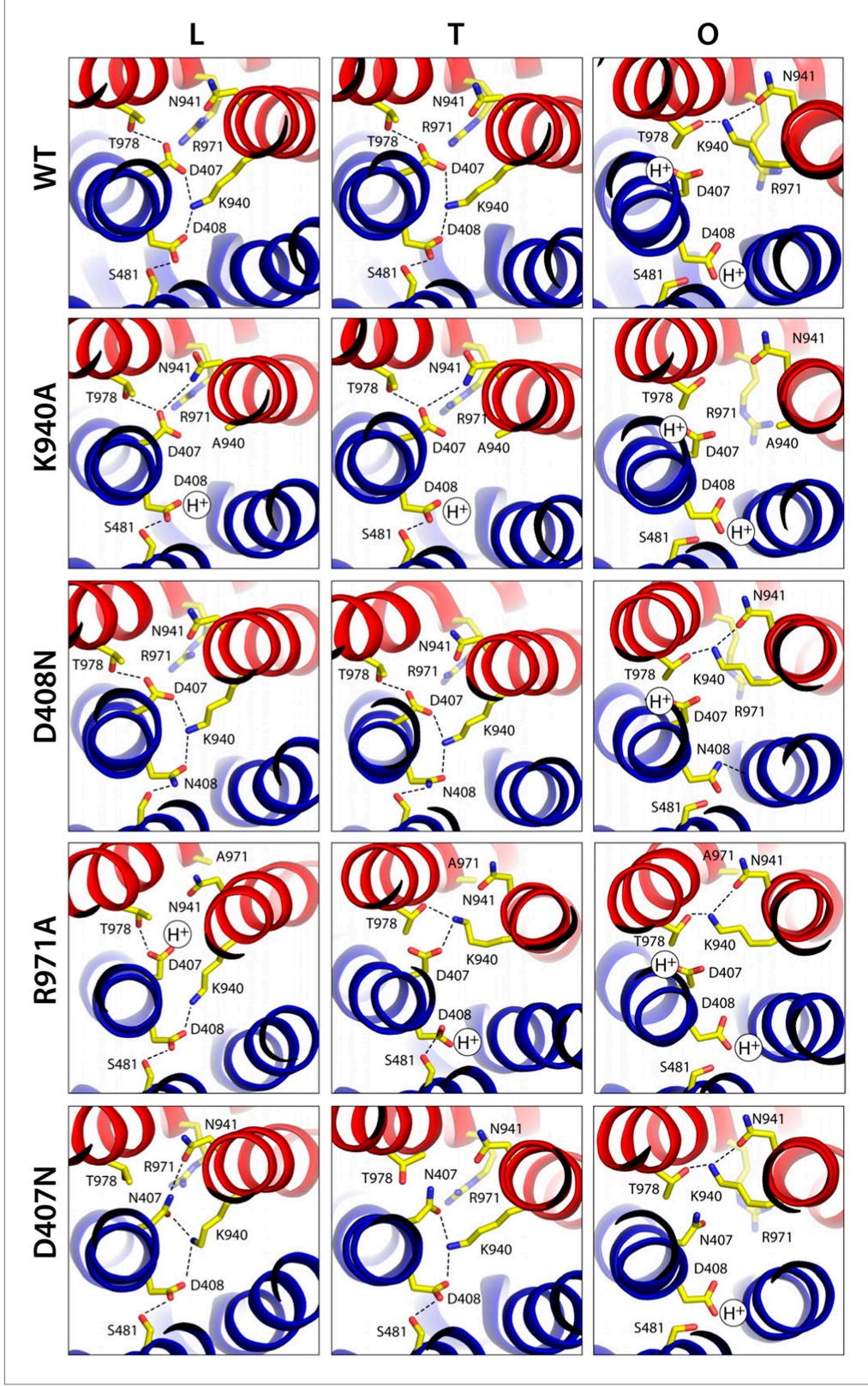

**Figure 7**. Configuration of the proton-binding site in asymmetric crystal structures of four AcrB variants inactive in drug efflux, compared to the wildtype protein. The binding site is viewed and represented as in *Figure 4*, omitting the electron-density map. The most probable protonation states of D407 and D408, based on Poisson–Boltzmann

*Figure 7. Continued on next page*

*Figure 7. Continued*

Monte-Carlo simulations (wildtype and mutants) as well as all-atom free-energy perturbation molecular dynamics simulations (wildtype), are indicated in each case with 'H+'. K940 and R971 are protonated in all cases, according to these calculations. Hydrogen-bonds identified on the basis of the distance and orientation of potential donors and acceptors, are indicated with dashed lines.

The following figure supplement is available for figure 7:

**Figure supplement 1**. Protonation probability (%) of selected side-chains in the transmembrane domain of AcrB, in crystal structures of wildtype and mutagenized variants of the transporter.

dependent on the conformational state of the protein, that is, they must be consistent with an alternating-access mechanism, so as to preclude leakage and ensure strict coupling of substrate and proton transport. To examine whether the conformational cycle of AcrB indeed implies changes in proton accessibility, we carried out an all-atom molecular dynamics simulation of the complete trimer in a lipid membrane, and analyzed the formation and persistence of hydration channels across the TM domain. The results, summarized in *Figure 9A*, indicate that the LTO conformational cycle described above does indeed involve a mechanism of alternating access. More specifically, we observe that the L state permits the formation of two persistent water channels open to the cytoplasm, at the interface between repeats R1 and R2, and reaching into the proton-binding site; access from the periplasmic space is, however, closed (*Figure 9A*). In the T state, by contrast, two narrow water wires form on the

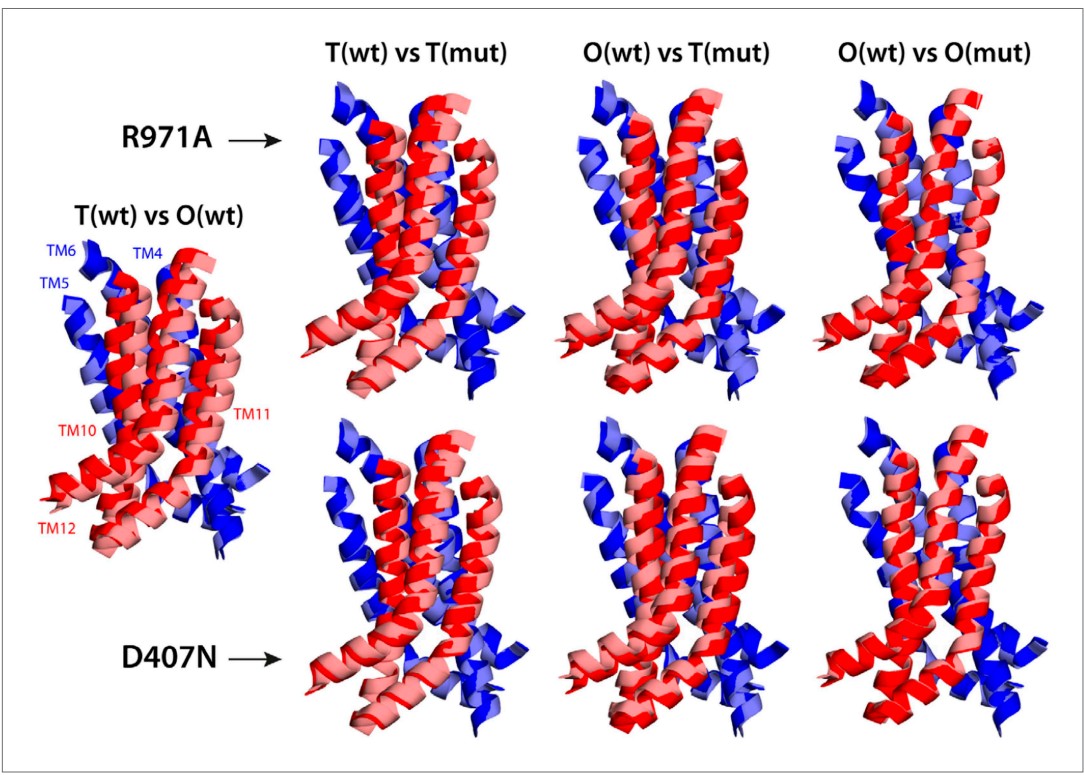

**Figure 8**. Relative motions of transmembrane repeats R1 and R2 in wildtype AcrB and in the R971A and D407N variants, in the T to O transition. For clarity, only the three transmembrane helices at the interface between repeats are shown, namely TM4–TM6 (R1, blue cartoons) and TM10–TM12 (R2, red cartoons), respectively. The structural superimpositions shown are optimal overlays of TM4–TM6, and thus highlight the differences in the orientation of TM10–TM12 relative to TM4–TM6. This comparison shows that the T state in the crystal structures of the R971A and D407N mutants is an intermediate between the T and O conformations in wildtype AcrB. The conformation of the O state, however, is largely unaffected by these mutations.

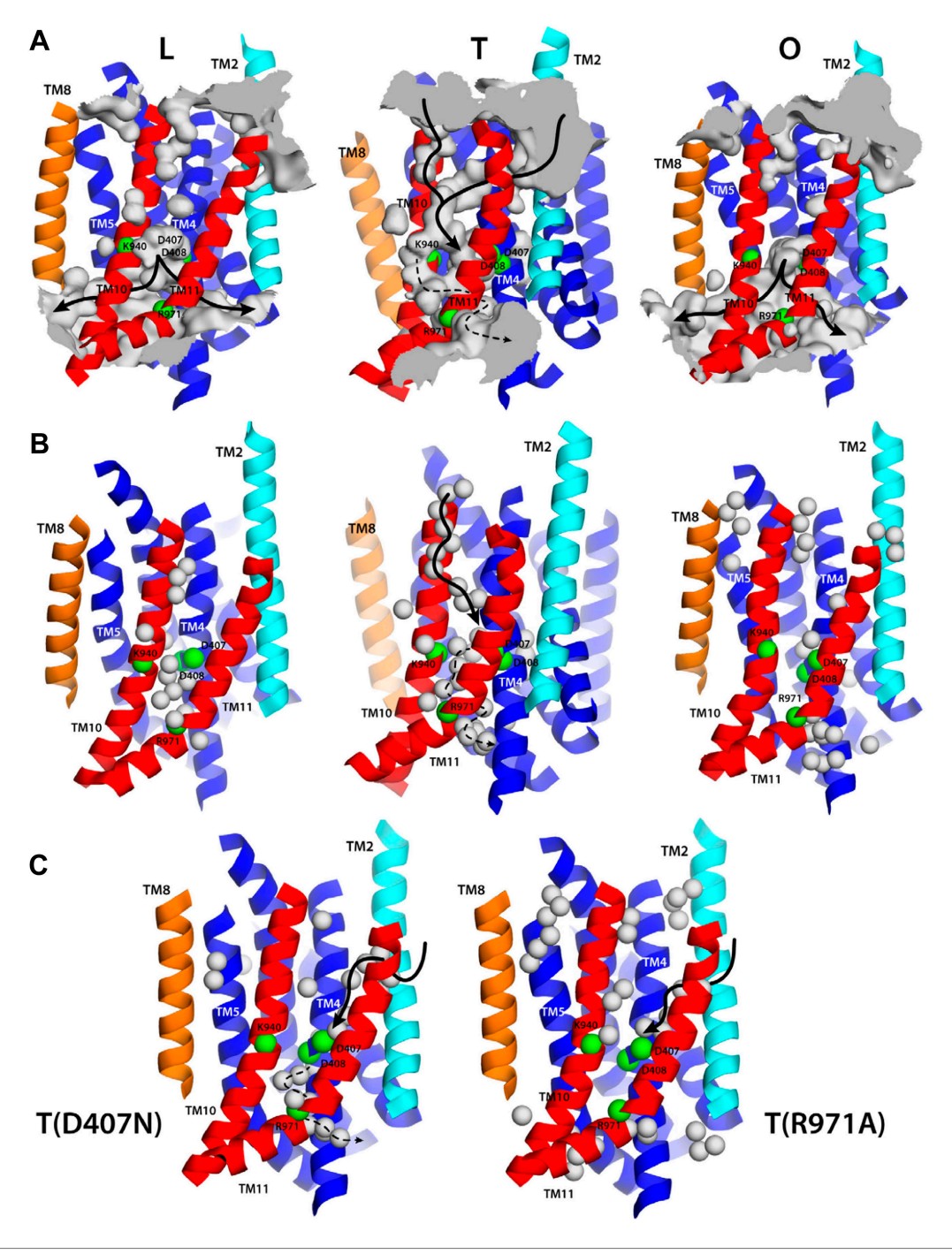

**Figure 9**. Alternating access to the proton-binding site in AcrB. (**A**) Conformation-dependent water channels within the TM domain of AcrB, observed in all-atom simulations in a phospholipid membrane. Transmembrane helices are represented as cartoons and colored as in *Figure 1*; for clarity, TM7, TM9 and TM12 in repeat R2 are omitted. The locations of D407, D408, K940, R971, E346 and D924 are indicated with green spheres representing their Cα atoms. The water channels are represented by average density maps computed from the molecular dynamics trajectory, and depicted as an iso-density surface (gray). Based on this analysis, the proton-binding site is open to the cytoplasm in the L and O states, via two possible channels (solid black lines), whereas in the T state it is open to the periplasm, also via two water wires (solid black lines). A third narrow wire in the T state seems to connect the proton-binding site to the cytoplasm (dashed black lines), but the positively charged R971, which traverses this water wire, likely blocks H⁺ leakage. (**B** and **C**) Crystallographic water molecules (gray spheres) detected within the TM domain of AcrB, in each of the protomer states of the wildtype structure, as well as in the T protomer of the D407N and R971A variants.

periplasmic face of the TM domain, again reaching into the proton-binding site, and approximately along the interface between the two repeats R1 and R2 (*Figure 9A*). Interestingly, E346 and D924 flank the entrance of each of these water wires; as mentioned earlier, both these side-chains are likely to be protonated in the L and T states and deprotonated in the O state (*Figure 4*, *Figure 5*), which suggests that E346 and D924 facilitate proton uptake by providing transient binding sites en route to D407/D408 (*Fischer and Kandt, 2011*). On the cytoplasmic side, we also observed a narrow water wire, flanked by R971, which is thus unlikely to sustain proton transfer, as discussed below. Finally, the O state features two water channels into the proton-binding site that are highly similar to those seen in the L conformation, that is, open to the cytoplasm, while access from the periplasmic space is also occluded (*Figure 9A*).

A striking feature of these putative access pathways for protons is that they are, for the most part, highly dynamic and permit individual water molecules to frequently exchange between the bulk solution and the close vicinity of the proton-binding site (*Figure 10A*). For the L state, for example, we observed more than 900 exchange events in a 250-ns period; about 70% occur via the cytoplasmic channel formed between helices TM5 (repeat R1) and TM10 (repeat R2), while the remainder proceeds via the channel formed between TM4 (R1) and TM11 (R2). Likewise for the O state, with over 600 exchange events in the same time-period, distributed over the two channels, roughly in a 2:1 ratio. In the T state, the periplasmic pathway flanked by TM2-TM4 and TM11 seems to be the most dynamic, although much less so than the cytoplasmic channels in L and O (we observed about 30 exchange events), while the periplasmic wire flanked by TM10 consists of a series of water molecules that are static on the simulation time-scale.

As mentioned, in the T state we also observed a significant number of water molecules exchanging between the proton-binding site and the cytoplasm. However, the trajectories followed by these water molecules invariably enter or exit the TM domain in close proximity to R971 (within 10 Å) (*Figure 10B*). This positively charged side-chain very likely imposes a very large energetic barrier for proton diffusion to and from the cytoplasm, despite the presence of water. Such an electrostatic barrier would be analogous, for example, to that observed in the selectivity filter of aquaporins, which is also lined by an arginine side-chain (*Kosinska Eriksson et al., 2013*). By contrast, almost half of the water-permeation events in the L and O states involve molecular trajectories that are distant from R971 at all times

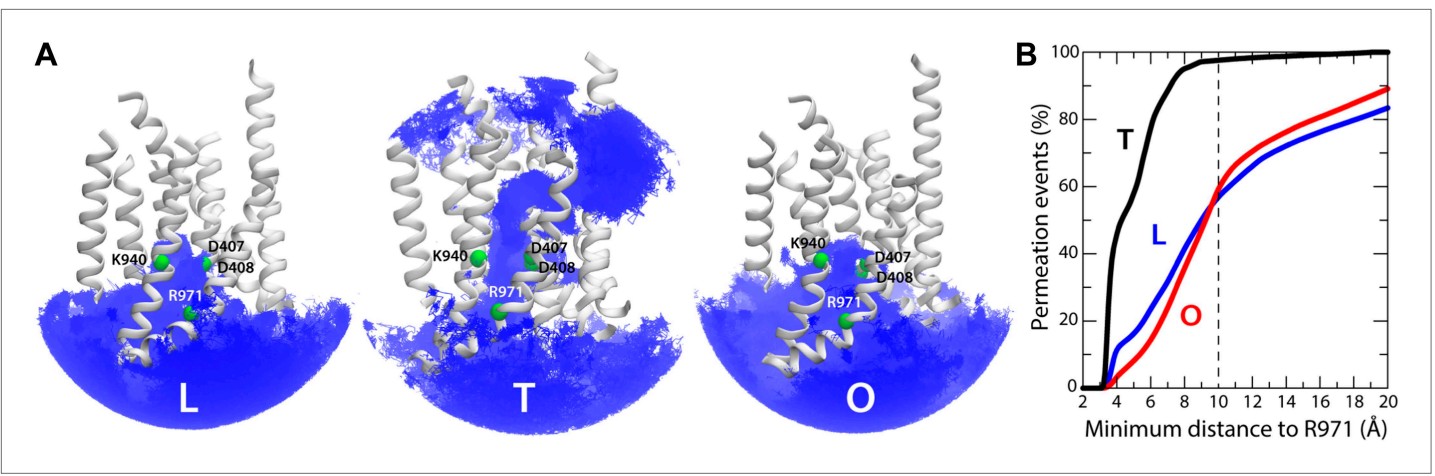

**Figure 10**. Dynamic exchange of water molecules between the proton-binding site in the core of the TM domain of AcrB and the bulk solution at either the periplasmic or cytoplasmic sides of the membrane and proposed role of R971. (**A**) A random selection of single-water trajectories extracted from the molecular dynamics simulation are shown as blue lines, cut-off at 30 Å from the proton-binding site. The location of residues D407, D408, K940 and R971 is indicated by their Cα atoms, shown as green spheres. Transmembrane helices in each protomer/state are represented as cartoons (white); for clarity, TM7, TM9 and TM12 in repeat R2 are omitted. (**B**) Proposed role of R971 as a selectivity filter against cytoplasmic H[+] leakage in the T state of AcrB. The plot quantifies the frequency of water exchanges between the proton-binding site within the TM domain of AcrB and the bulk solution at the cytoplasmic side of the membrane, as a function of the proximity of the individual molecular trajectories to R971. In the T state, almost 100% of the water trajectories approach R971 within 10 Å, and therefore this water wire (dashed lines in *Figure 9*) is highly unlikely to sustain H[+] conduction, for electrostatic reasons. In the L and O states, by contrast, 40% of the exchanging water molecules follow trajectories that do not approach R971 (solid lines in *Figure 9*).

(*Figure 10B*). Likewise, no large electrostatic barriers for protons are apparent along the periplasmic pathways in the T state.

That the hydration of the TM domain is conformationally dependent is entirely consistent with the electron densities from ordered water molecules in the X-ray structures of wildtype and variant AcrB. As shown in *Figure 9B*, the structure of the T state in wildtype AcrB features the abovementioned continuous water wire flanked by TM10, connecting the periplasmic space and D407/D408, with D924 at its entrance. The second outward-open wire, with E346 at its entrance and running along the interface between TM2-TM4 and TM11, cannot be discerned in the wildtype structure, but interestingly, is partially resolved in the structures of D407N and R971A (*Figure 9C*). As discussed earlier, in these crystals the T protomer is captured in an intermediate conformation between wildtype T and O, in which the lateral displacement of repeat R2 is already underway (*Figure 8*). In the native membrane and at room temperature, the two repeats would be much more dynamic than what appears in the crystal lattice, and thus we envisage that both water wires co-exist, also in the wildtype protein, as we observe in the simulation. In any case, it is apparent that no water wires connect the central proton-binding site and the periplasmic space in the L and O states, in any of the structures. By contrast, a significant number of ordered water molecules are observed in the intracellular side of these protomers. According to our simulation, the cytoplasmic channels, and particularly that between TM5 and TM10, are highly dynamic, and thus electron densities in this region would be poorly defined. Nevertheless, ordered water can be clearly discerned in the proximity of the proton-binding site, particularly in the L state (*Figure 9B*).

In sum, our data indicate that throughout the conformational cycle of AcrB the proton-binding sites within the TM domain become alternately accessible to the periplasm or the cytoplasm. As mentioned, this is a necessary condition in order for the transporter to harness the proton-motive-force. Interestingly, two distinct types of relative motion of repeats R1 and R2 result in alternating access, namely a rocking motion during the L to T transition, with no protons bound, and a shear motion in the T to O transition, in the protonated form. This finding indicates that other transporters may also follow significantly different conformational pathways in the substrate-bound and apo segments of their functional cycle, contrary to what is often assumed. To our knowledge, however, AcrB is the only secondary-transporter for which two distinct structural transitions resulting in alternating access have been described to date.

## The antiport mechanism of AcrB—conformational coupling within and across protomers

As described above, AcrB consists of well-defined structural elements in both its periplasmic and membrane domains, whose relative arrangement changes to a significant degree throughout the conformational cycle of the transporter. In the periplasmic domain, these rearrangements reflect substrate loading from the periplasm, the re-shaping of the hydrophobic binding site, and its release into the TolC lumen. In the membrane domain, water pathways across the protein become reconfigured to facilitate access from either side of the membrane, and a network of ionic and H-bonding interactions is re-organized as a result of protonation and deprotonation events.

Taken together, these observations reveal an unprecedented antiport strategy that involves the remote coupling of two alternating-access transport mechanisms, that is, one for drugs and another for protons, within each of the AcrB protomers, which we summarize in *Figure 11*. In the L to T transition, substrate binding in the deep binding pocket of the porter domain causes a structural change within the PN2/PC1 unit. This change translates into the downward displacement of TM2, which pushes on Iα and causes it to swing away, along with whole of the R2 repeat (*Figure 3*). In this transition, the interaction network formed by the proton-relay triad, D407, D408 and K940, remains engaged (*Figure 4*), with all three side-chains ionized (*Figure 5*, *Figure 6*), and seems to provide the hinge point of the swinging motion of R1 and R2 relative to each other (*Figure 3*). Importantly, this motion leads to alternating access of pathways for protons from either side of the membrane. Specifically, while in the L state the triad is accessible only from the cytoplasm, in the T state protons can only enter the TM domain from the periplasm (*Figure 9*, *Figure 10*). In both cases, proton transfer is highly likely to be mediated by water channels formed at the interface between repeats, which open or close as the repeats themselves rearrange relative to each other.

In the T to O transition, D407, D408 and K940 break away from each other (*Figure 4*), following binding of one proton to each of the aspartate side-chains (*Figure 5*, *Figure 6*). K940 remains in the

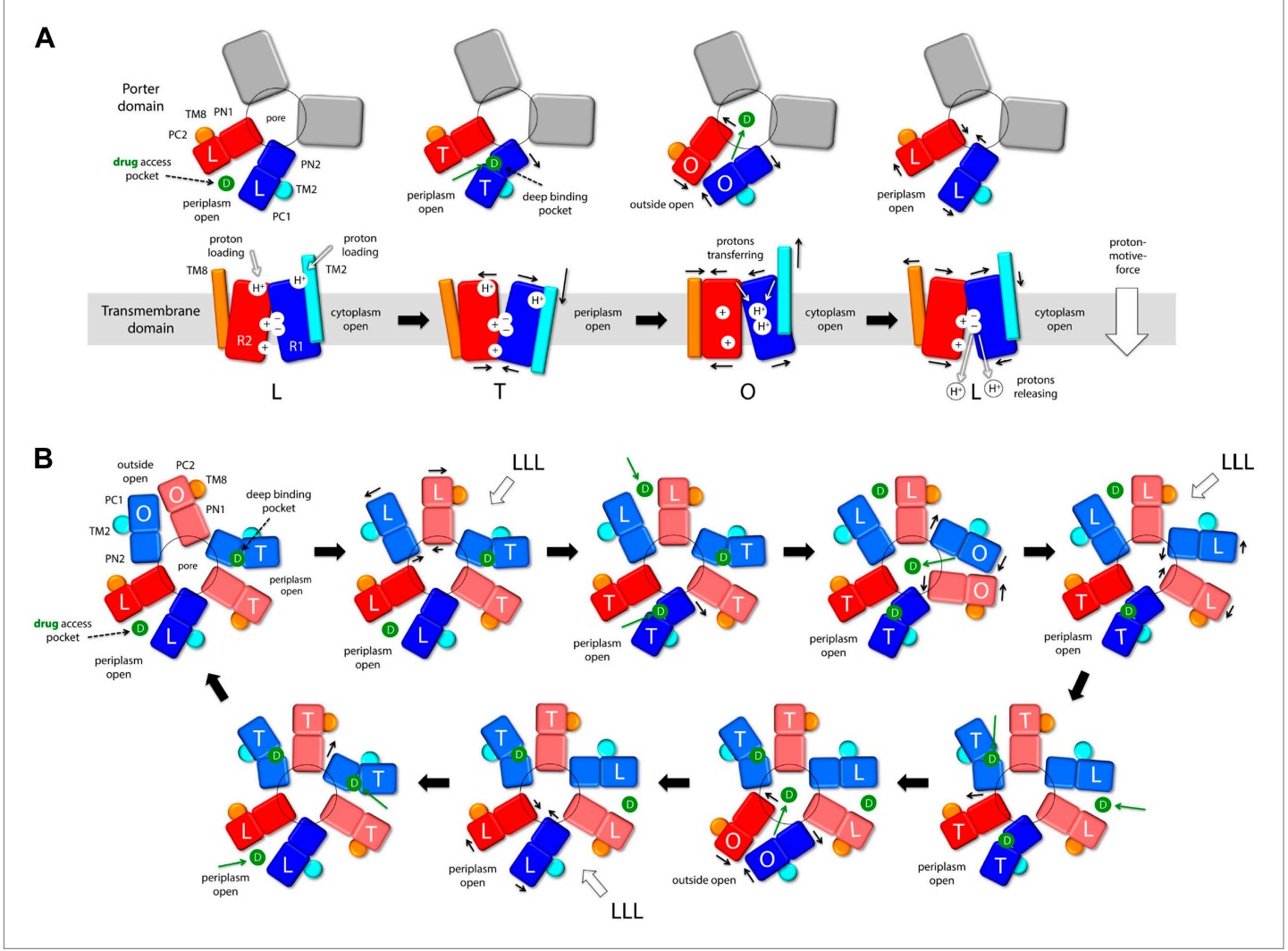

**Figure 11**. Proposed mechanism of drug-proton antiport in AcrB. (**A**) The scheme shows the conformational cycle of one of the protomers; the porter domain is viewed from the outer membrane, and the TM domain is viewed along the membrane plane. The structural repeats in the TM and porter domains, R1 and R2, and PC1/PN2 and PN1/PC2, respectively, are colored as in *Figure 1*. Likewise for the TM2 and TM8, which flank R1 and R2 and couple them to the repeats in the porter domain. The conformational state (L/T/O) of each protomer is indicated alongside the proposed location of the substrate (D) and the protons (H⁺) in each state. Residues in the proton relay network formed by K940, R971, D407 and D408, which controls the relative orientation of R1 and R2, are indicated with '+' and '−' symbols. The series of states proposed for the TM domain is based on the analysis of protonation states and water accessibility of wildtype AcrB and its variants, reported above. The model postulates that TM2 and TM8 mediate the strict coupling between these two conformational cycles. (**B**) The scheme shows the conformational cycle of the porter-domain trimer, viewed from the outer-membrane. The specific series of states proposed for the porter domain derives from analysis of potential steric clashes at the protomer–protomer interface in all possible combinations of L, T and O conformations (*Figure 11—figure supplement 1*). Note that the frequency of protomers in the O state is minimized, relative to other possible cycles, while that of protomers in the L and T state is maximized; we propose this kinetically advantageous, in that it would result in minimum back-flow of substrates from the exit pore, and maximal capture from the periplasmic space.

The following figure supplement is available for figure 11:

**Figure supplement 1**. Mechanistic significance of the conformational coupling among protomers.

ionized, protonated state, and retracts towards TM10/TM11 to compensate for the loss of interaction partners. R971 (on TM11) also reconfigures in response to these events, rotating towards TM10. As a result, the repeats R1 and R2 become disengaged, and R2 (along with Iα) moves laterally relative to R1 (*Figure 3*). This displacement allows TM2 to return to the up conformation, and importantly, enables TM8 to come into close proximity to R2. The displacement of TM8 causes the PN1/PC2 repeat in the

porter domain to move, as a rigid body, against PN2/PC1, thus closing the binding cleft to the periplasmic space, and triggers instead the opening of an exit pathway towards the funnel domain. The ability of TM8 to steer the motion of PN1/PC2 in the T to O step (*Video 5*) seems to be facilitated by the elongation of the helical fold of TM8, on the periplasmic side, upon transition from L to T (*Video 4*); this naturally facilitates its coupling to the motion of PN1/PC2. Crucially, in the O state the proton-relay site is again accessible to the cytoplasm, and not to the periplasm (*Figure 9*, *Figure 10*). Thus, deprotonation will imply a release of the bound protons down the electrochemical gradient. Proton release into the cell interior also causes the R1 and R2 repeats to re-engage (*Figure 4*), and thus the dissociation of TM8 from R2 when reforming the L state (*Figure 3*). This enables the re-opening of the drug-binding cleft between PC1 and PC2, exposing the access binding pocket (*Nakashima et al., 2011*; *Eicher et al., 2012*) to the periplasm, ready to sequester a new drug molecule.

It is worth noting that the directionality of this mechanism is entirely encoded structurally, for each of the protomers. Proton uptake leads to substrate export (instead of symport) for two reasons: first, both proton- and substrate-binding sites are concurrently open to the periplasm in the T state, while in the O state they become open to the cytoplasm and the TolC lumen, respectively; second, both porter and membrane domains are able to re-open to the periplasm after releasing their cargo, that is, in their apo state (*Figure 11A*, drug access pocket and proton loading). Thus, the free-energy gain associated with downhill proton permeation compensates the cost of uphill efflux of cytotoxic compounds. This energy-transduction process is reminiscent of the binding-change mechanism of the $F_1F_o$-ATP synthase, which also utilizes the energy harnessed from the transmembrane electrochemical proton gradient to release synthesized ATP molecules from their binding sites in the α/β subunits (*Boyer, 1997*; *Pos, 2009*).

Although the mechanism and energetics of transport in AcrB can be fully rationalized in terms of the individual protomers, the extensive trimer interface within the porter domain suggests that the protomers are conformationally coupled. Indeed, transport of certain cephalosporins and penicillins has been shown to be highly cooperative (*Nagano and Nikaido, 2009*; *Lim and Nikaido, 2010*). How the cross-talk among protomers might benefit the antiport mechanism is, however, not immediately evident. From an architectural standpoint, trimerization is necessary for AcrB to form a pore leading into the TolC channel, but why would the protomers not cycle independently from each other? A disadvantage of independent protomer cycling would be that a drug released into the exit pore by a protomer in the O state would often be exposed to a second protomer also in the O state, which would enhance the probability of re-binding. Such event would not likely result in drug re-uptake (which would require uphill proton transport) but efflux would be surely hindered from a kinetic standpoint, because the drug concentration within TolC is probably very large. Ideally, therefore, the number of trimer configurations in which one or more protomer are in the O state ought to be somehow minimized. Following the same kinetic argument, it would be advantageous if AcrB maximizes the number of trimer configurations featuring the L and T states in one or more protomers, so as to accelerate capture (or re-capture) of the substrate from the periplasmic space or the inner membrane.

Remarkably, analysis of computer-generated models of the porter domain in all non-symmetric permutations of the L, T and O states ('Materials and methods') indicates that this kind of kinetic optimization is what emerges from the conformational coupling among protomers. A quantification of the steric clashes between protomers in these hypothetical models indicates that OO and LO interfaces (counter-clockwise, viewed from TolC) are highly incompatible, and that OT is also suboptimal. Thus, most of the combinations that include one or more O states in the trimer would be energetically unfeasible (*Figure 11—figure supplement 1*). *Figure 11* depicts the conformational cycle of the porter domain that involves only the most plausible combinations, alongside the cycle of the TM domain described earlier. Throughout this cycle, the proportion of protomers in either the L or T state relative to those in the O state is fourfold greater than in a hypothetical uncoupled trimer (*Figure 11—figure supplement 1*). That is, the AcrB protomers are substantially more efficient efflux pumps if they are coupled; not because the protomers energize each other, but because collectively they are able to restrict their conformational pathways so as to achieve the most productive mechanism.

## Conclusions

New high-resolution crystal structures of wildtype and AcrB variants, systematically analyzed through diverse computational methods, have enabled us to formulate a novel, clear-cut theory of the

mechanism of proton-coupled drug-efflux in this very complex transport system. On the basis of this theory, which is consistent with existing biochemical, structural and functional data, additional and more precise experiments and computations may now be designed to further establish the proposed molecular mechanism of transport and energy transduction. It will also be of interest to extrapolate these findings to other homologues in the RND superfamily. Such investigations could potentially aid the development of novel pharmacological approaches, such as conformational inhibitors, against Gram-negative multidrug resistant pathogens.

## Materials and methods

### Bacterial strains, plasmids and growth conditions

*E. coli* C43 (DE3) (*Miroux and Walker, 1996*) harboring mutant derivatives of pET24acrB$_{His}$ (*Pos and Diederichs, 2002*; *Seeger et al., 2009*) were used for protein overexpression. LB medium and LB agar (*Sambrook et al., 1989*) were used for routine bacterial growth at 37°C. Terrific broth (*Tartoff and Hobbs, 1987*) was used for protein overexpression. Kanamycin (Applichem, Germany) was applied at 50 μg ml$^{-1}$ (Kan$^{50}$).

### Crystallization of variant AcrB in complex with DARPins

Overexpression and membrane preparation of wildtype and mutant AcrB was carried out as described previously (*Pos and Diederichs, 2002*; *Seeger et al., 2006*). DARPin clone 1108_19 (*Sennhauser et al., 2007*) was overexpressed and purified according to *Binz et al. (2003)*. To crystallize the AcrB/DARPin complex, the method described by *Sennhauser et al. (2007)* with *n*-dodecyl-β-D-maltoside as detergent (Glycon Biochemicals, Germany) was applied. For all AcrB variants except R971A, minocycline was added prior to crystallization at a concentration of 2 mM (Sigma-Aldrich, Germany). All crystals were grown by the hanging drop vapor diffusion method at 0.05 M ADA buffer pH 6.5, 7–9% polyethylene glycol (PEG) 4000, 6–10% glycerol, 0.2 M (NH$_4$)SO$_4$. Cryprotection was achieved by transfer of the crystals to the same buffer containing 30% glycerol and flash freezing into liquid N$_2$.

### X-ray diffraction dataset analysis and refinement procedure

Datasets of P2$_1$2$_1$2$_1$ crystals were collected at beamline X06SA (wavelength 0.8–1.0 Å) of the Swiss Light Source (Paul Scherrer Institut, Villigen, Switzerland). Data reduction was done with the XDS package (*Kabsch, 2010*). The structures were solved by molecular replacement using MOLREP (*Vagin and Teplyakov, 2010*) or PHASER (*McCoy et al., 2007*). As search model pdb entry 4DX5 (*Eicher et al., 2012*) was used and model rebuilding was employed using COOT (*Emsley et al., 2010*). Refinement was carried out with PHENIX (*Adams et al., 2010*) and REFMAC5 (*Murshudov et al., 2011*) and additional water molecules appended. Model rebuilding and water molecule analysis was done employing COOT (*Emsley et al., 2010*). Figures were created using PyMOL (The PyMOL Molecular Graphics System, Version 1.5.0.3 Schrödinger LLC, Oregon, USA).

### Calculation of protonation probabilities

Protonation probabilities were calculated using two independent methodologies. Wildtype and mutagenized AcrB structures were analyzed via Monte-Carlo/Poisson-equation simulations of the global protonation equilibrium in the complete trimer. In addition, selected ionizable side-chains in the wildtype transporter were analyzed via all-atom molecular dynamics simulations of individual protomers, using the free-energy perturbation (FEP) method. To assess the most likely protonation state of the ionizable side-chains in AcrB, we carried out a series of Metropolis Monte-Carlo simulations of the global protonation equilibrium of the protein at fixed pH. In a first set of calculations, we analyzed in each protomer separately, including all ionizable side-chains in the calculation. All lysine and arginine side-chains were consistently predicted to be protonated at the pH and dielectric constant values used in these calculations (see below). Thus, in subsequent simulations, in which the complete trimer was considered, all lysine and arginine side-chains were fixed in their ionized state, except for K940 and R971, which were explicitly re-analyzed along with all aspartate, histidine and glutamate residues (306 side-chains in total).

   Each Monte Carlo simulation was initiated with a random configuration of protonation states, and consists of five cycles of temperature annealing and a subsequent equilibration. In the annealing phases, the temperature was raised to 400 K and then gradually reduced to 300 K over 100,000 simulation steps. The equilibration phases, in which the temperature was kept at 300 K, lasted 500,000 steps. The

protonation probability of each group was determined by averaging the ionisation states sampled during the equilibration phase of the final cycle of the simulation. This procedure was repeated 10 times with independent starting configurations, resulting in a global average; the typical statistical error was <2%.

To evaluate the energy changes associated with the Monte Carlo moves we used a continuum electrostatic model, that is, the crystal structure of the AcrB trimer is surrounded by a high-dielectric medium representing water, except for a low-dielectric rectangular slab surrounding the transmembrane domain, which mimics the lipid membrane. Specifically, we used the linearized Poisson-Boltzmann equation solver implemented in CHARMM c34a2 (**Brooks et al., 2009**) to set up the continuum model and to compute all the necessary electrostatic energy contributions. In all simulations the dielectric constant of the solvent was 80, and that of the membrane was 2. The results described in the manuscript and represented in **Figure 5** and **Figure 6** correspond to a protein dielectric constant of 4; alternative values ranging from 2 to 6 were also explored, which resulted in analogous conclusions (not shown). The atomic charges employed are those in the CHARMM27 force field (**MacKerell et al., 1998**). The atomic radii are those previously optimized for continuum electrostatic calculations based on the CHARMM27 force field (**Nina et al., 1997**). Non-standard radii were adapted according to chemical similarity, for example, the radius of the N$\zeta$ atom in the neutral lysine was considered to be equal to that of the side-chain nitrogen atom of asparagine/glutamine.

The reference aqueous-phase p$K_a$ values were: aspartate, 4.0; glutamate, 4.4, histidine, 6.3; lysine, 10.4; and arginine, 12.0. Residues on the periplasmic side of the protein were assumed to exchange protons with a solution at pH$_{out}$ = 5.5, while those on the cytoplasmic side exchanged with a solution at pH$_{in}$ = 7.5; interpolated pH values in this range were used for residues across the membrane. Calculations with an alternative value of pH$_{out}$ = pH$_{in}$ = 7.5 led to analogous conclusions (not shown).

## Molecular dynamics simulations

All molecular dynamics simulations were carried out at 298 K and 1 atm. Three independent all-atom systems were studied, namely the complete asymmetric AcrB trimer and truncated constructs of the T and O protomers comprising the transmembrane and porter domains only (**Figure 1—figure supplement 2**). In all cases the protein structures were embedded in a hydrated POPC bilayer using GRIFFIN (**Staritzbichler et al., 2011**). All simulations were initiated with the high-resolution (1.9 Å) crystal structure of wildtype AcrB (PDB entry 4DX5). The simulation system for the complete AcrB trimer includes ~369,000 atoms, enclosed in a periodic orthorhombic box of dimensions ~150 × 150 × 160 Å. This system was simulated for 250 ns (after equilibration) to assess the formation and dynamics of water channels within the transmembrane domain. The protomer systems include ~101,000 atoms in a periodic orthorhombic box of dimensions ~90 × 90 × 120 Å. These systems were used to evaluate the most likely protonation state of selected side-chains using the FEP method. Each of these calculations entailed a total of ~80 ns of simulation (after equilibration). Three independent all-atom simulation systems were studied, namely the complete asymmetric AcrB trimer (residues 1 to 1044 in each protomer) (**Figure 1—figure supplement 2**); a truncated constructs of the T protomer, comprising the transmembrane and porter domains only (residues 1 to 182, 275 to 723 and 813 to 1044) (**Figure 1—figure supplement 2**); and an analogous truncated construct of the O protomer. In all cases the protein structures were embedded in a hydrated phospholipid (POPC) bilayer using the GRIFFIN method (**Staritzbichler et al., 2011**), adding Na$^+$ counter ions when necessary to neutralize the total charge of the system.

This system was employed to assess the formation and dynamics of water channels within the transmembrane domain. A minocycline molecule was bound to the porter domain in the T protomer. D407 and D408 were deprotonated in the L and T protomers, and protonated in the O protomer; K940 and R971 were protonated in all protomers. The simulation was initiated with the high-resolution (1.9 Å) crystal structure of wildtype AcrB (PDB entry 4DX5), preserving crystallographic water molecules. Additional water molecules within the protein structure were placed with DOWSER (http://danger.med.unc.edu/hermans/dowser/dowser.htm). To optimize the protein-solvent and protein-lipid interfaces in the model, and to thermalize the system, a series of short simulations (10 ns) were initially carried out with gradually weaker positional restraints applied to the protein and the crystallographic and DOWSER-added water molecules (excluding hydrogen atoms). To enable the protein to tumble, the equilibration phase was extended (5 ns) after replacing the positional restraints with collective conformational restraints (RMSD), applied first to the whole protein, and subsequently to each of three transmembrane domains separately. In the last state of the equilibration

(10 ns), only the configuration of the proton-relay network was preserved via inter-residue distance restraints (flat-bottom harmonic potential). Finally, a fully unrestrained simulation was carried out for 250 ns.

The truncated-protomer systems were used to re-evaluate the most likely protonation state of selected side-chains using the free-energy perturbation (FEP) method; the substantial computational cost of this methodology is what motivated the truncation of the protein. The preparation and equilibration of these reduced simulation systems was analogous to the procedure followed for the trimer. In the final production phase (90 ns), however, only the transmembrane domain was fully unrestrained. The conformation of the porter domain was preserved using a harmonic restraint on its RMSD relative to the crystal structure, to preclude any distortions resulting from the artefactual truncation of the funnel domain. Importantly, the force constant of this harmonic restraint was selected so that the structural fluctuations of the porter domain in the truncated-protomer system are comparable to those observed in the unrestrained simulations of the complete trimer (not shown). Thus, the dynamics of the transmembrane domain during the free-energy calculations is unlikely to be significantly affected by the truncation. For both T and O protomers, snapshots after 90 ns of simulation were extracted and used to initiate the FEP calculations. Here, the protonation state of a given side-chain is exchanged alchemically, using a step-wise protocol controlled by a parameter $\lambda$ that reflects the weight of either protonation state in the potential energy function of the system. All FEP calculations were carried out in the forward and backward directions, with 38 intermediate $\lambda$ steps (33 for D408 in the O state). The simulation time per $\lambda$ step was 1 ns; the initial 100 ps of each simulation were considered as equilibration, and so free-energy increments were obtained from the final 900 ps.

All molecular dynamics simulations were carried out with NAMD 2.7 (*Phillips et al., 2005*), using an integration time-step of 2 fs, periodic boundary conditions, and a Langevin thermostat set to 298 K. The pressure along the direction perpendicular to the membrane was maintained at 1 atm using a Nose-Hoover Langevin piston, while keeping the surface area of the membrane constant (~69 Å$^2$ per lipid). Electrostatic interactions were calculated using the Particle-Mesh-Ewald algorithm, with a real-space cut-off of 12 Å. A shifted Lennard-Jones potential, also cut-off at 12 Å, was used to compute van-der-Waals interactions. The CHARMM27/CMAP force field for proteins and lipids (*MacKerell et al., 1998*, *2004*) was used in all calculations. Force field parameters for minocycline were taken from *Aleksandrov and Simonson (2009)*. Water density maps were calculated with VMD (*Humphrey et al., 1996*).

## Statistical analysis of lysine interactions in the PDB

The frequency of hydrogen-bonded interaction networks involving lysine side-chains and other polar and/or acidic residues was quantified using a subset of entries in the Protein Data Bank that consists of all protein X-ray structures of resolution 3.0 Å or better, excluding redundant entries with sequence identity greater than either 30% or 70%. A distance threshold of 3.0 Å was used to identify hydrogen-bonds donated by lysine N$\zeta$ atoms to potential acceptor oxygen atoms in either Asp (O$\delta$), Asn (O$\delta$), Glu (O$\epsilon$), Gln (O$\epsilon$), Ser (O$\gamma$) or Thr (O$\gamma$) side-chains.

## Steric exclusions in the conformational cycle of the porter domain

Alternative structures of the porter domain were modeled in all possible permutations of the L, T and O conformations, via rigid-body rotations of the PC1/PN2 and PN1/PC2 sub-domains around the trimer axis. All side-chains in all models (including the crystallographic LTO state) were then re-built using SCWRL4 (*Krivov et al., 2009*). The solvent-excluded surface (probe radius of 1.4 Å) of each protomer in each model was then calculated with MSMS (*Sanner et al., 1996*), and mapped onto a fine cubic grid. The overlap volume at each of the three promoter interfaces was computed by counting the number of grid-points concurrently inside two opposing surfaces. The models were ranked according to the total value of the overlapping volume.

## Acknowledgements

This work was supported by funds from the Swiss National Foundation (KMP); the German Research Foundation Collaborative Center SFB807 (KMP); the German Research Foundation Cluster of Excellence EXC115 (KMP and JDF-G); the Division of Intramural Research at the National Heart, Lung & Blood Institute of the National Institutes of Health (JDF-G); and by studentships from the University of Zurich (LB), the University of Konstanz (WZ) and the Chinese Scholarship Council (WZ). The research leading

to these results was conducted as part of the Translocation consortium (www.translocation.eu) and has received support from the Innovative Medicines Joint Undertaking under Grant Agreement n°115525, resources which are composed of financial contribution from the European Union seventh framework program (FP7/2007-2013) and EFPIA companies in kind contribution. We are also thankful to the beamline staff at the Swiss Light Source (PXI, SLS) of the Paul Scherrer Institut in Villigen (Switzerland). We also thank Fabrizio Marinelli for his assistance with the PDB statistical analysis, and Lucy Forrest for her comments on this manuscript.

Atomic coordinates and structure factors for the reported crystal structures are deposited in the Protein Data Bank under accession codes 4U8V (AcrB_D407N), 4U8Y (AcrB_D408N), 4U95 (AcrB_K940A) and 4U96 (AcrB_R971A).

## Additional information

### Funding

| Funder | Grant reference number | Author |
| --- | --- | --- |
| Deutsche Forschungsgemeinschaft | Cluster of Excellence DFG-EXEC115 | Klaas M Pos, José D Faraldo-Gómez |
| Swiss National Science Foundation | 31003A_118402 | Klaas M Pos |
| German Research Foundation Collaborative Research Centre | DFG-SFB807_TP18 | Klaas M Pos |
| Innovative Medicines Initiative Joint Undertaking Project Translocation | GA_115525 | Klaas M Pos |
| National Heart, Lung, and Blood Institute | Division of Intramural research | José D Faraldo-Gómez |
| University of Zurich | | Lorenz Brandstätter |
| University of Konstanz | | Wenchang Zhou |
| Chinese Scholarship Council | CSC2008101067 | Wenchang Zhou |
| Swiss National Science Foundation | Swiss National Science Foundation Professorship PP00P3_144823 | Markus A Seeger |

The funders had no role in study design, data collection and interpretation, or the decision to submit the work for publication.

### Author contributions

TE, MAS, JDF-G, KMP, CA, Conception and design, Acquisition of data, Analysis and interpretation of data, Drafting or revising the article; WZ, Acquisition of data, Analysis and interpretation of data, Drafting or revising the article; LB, Acquisition of data, Drafting or revising the article; FV, KD, Drafting or revising the article, Contributed unpublished essential data or reagents

## Additional files

### Supplementary files

• Supplementary file 1. Structural variations in the porter domain of the periplasmic region of wildtype AcrB, in the L, T and O states.

• Supplementary file 2. Data collection and refinement statistics for the crystal structures of AcrB mutants D407N, D408N, R971A, and K940A.

• Supplementary file 3. Differences in the structures of the repeats in the transmembrane and porter domains of AcrB, in wildtype vs. D407N, D408N, R971A, and K940A variants.

## Major datasets

The following datasets were generated:

| Author(s) | Year | Dataset title | Dataset ID and/or URL | Database, license, and accessibility information |
|---|---|---|---|---|
| Eicher T, Seeger MA, Anselmi C, Zhou W, Brandstätter L, Verrey F, Diederichs K, Faraldo-Gómez JD, Pos KM | 2014 | AcrB_D407N | http://www.pdb.org/pdb/explore/explore.do?structureId=4U8V | Publicly available at RCSB Protein Data Bank. |
| Eicher T, Seeger MA, Anselmi C, Zhou W, Brandstätter L, Verrey F, Diederichs K, Faraldo-Gómez JD, Pos KM | 2014 | AcrB_D408N | http://www.pdb.org/pdb/explore/explore.do?structureId=4U8Y | Publicly available at RCSB Protein Data Bank. |
| Eicher T, Seeger MA, Anselmi C, Zhou W, Brandstätter L, Verrey F, Diederichs K, Faraldo-Gómez JD, Pos KM | 2014 | AcrB_K940A | http://www.pdb.org/pdb/explore/explore.do?structureId=4U95 | Publicly available at RCSB Protein Data Bank. |
| Eicher T, Seeger MA, Anselmi C, Zhou W, Brandstätter L, Verrey F, Diederichs K, Faraldo-Gómez JD, Pos KM | 2014 | AcrB_R971A | http://www.pdb.org/pdb/explore/explore.do?structureId=4U96 | Publicly available at RCSB Protein Data Bank. |

The following previously published dataset was used:

| Author(s) | Year | Dataset title | Dataset ID and/or URL | Database, license, and accessibility information |
|---|---|---|---|---|
| Eicher T, Cha HJ, Seeger MA, Brandstatter L, El-Delik J, Bohnert JA, Kern WV, Verrey F, Grutter MG, Diederichs K, Pos KM | 2012 | Transport of drugs by the multidrug transporter AcrB involves an access and a deep binding pocket that are separated by a switch-loop | http://www.pdb.org/pdb/explore/explore.do?structureId=4dx5 | Publicly available at RCSB Protein Data Bank. |

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
