## [Decision Letter]

Thank you for sending your work entitled “Coupling of remote alternating-access transport mechanisms for protons and substrates in the multidrug efflux pump AcrB” for consideration at *eLife.* Your article has been favorably evaluated by Michael Marletta (Senior editor), a Reviewing editor, and 2 reviewers, both of whom (Ben Luisi and Shimon Schuldiner) have agreed to reveal their identity.

The Reviewing editor and the other reviewers discussed their comments before we reached this decision, and the Reviewing editor has assembled the following comments to help you prepare a revised submission.

Both reviewers are in agreement that your work should be accepted pending minor changes.

*Reviewer #1 minor comments*:

1) The movies are wonderful, but it would be easier to follow them if they were annotated with some labels.

2) In the Introduction the authors refer to the membrane distal portion of the periplasmic domain as the 'TolC-dockign domain'. Is this naming convention too suggestive in light of the report of the structure of the pump assembly (see [8] Nature)?

3) Figure 4 is a new interpretation of old data. The density looks a little poor in the L state: are the side chains disordered?

4) Abstract, change to read “...one of the most...”

*Reviewer #2 minor comments*:

1) What do the authors mean by saying that the ...TM domain had not previously described…?

2) Figure 5: there is an isolated residue in the cytoplasmic side that changes its protonation probability during the cycle. This should be labeled.

3) Figure 10 legend. They refer to figure dashed lines and solid lines in Figure 8, should be Figure 9.

4) Regarding the suggested role of R971 in blocking proton leakage: is there any indication that R971A expression is toxic to the cell?

5) Are the mutants D407N and D408N fully inactive? Could it be that they are moving only one proton at a time and they can then generate smaller gradients? Is it possible to look at AcrB-dependent passive loading of ethidium in whole cells? Is the assay that follows hydrolysis of cephalosporins sensitive enough for this purpose?

6) The model presented here has some very clear predictions that could be challenged experimentally or supported by existing data. Is there any evidence that supports a stoichiometry of 2H+/substrate? As far as I know AcrB transports positively and negatively charged substrates as well as neutral ones. The driving force for them with this stoichiometry would be very different at different pHs. I am aware of the problem of making proteoliposomes that transport substrate but they may be able to make PLs and follow substrate-induced proton transport.

---

## [Author Response]

Reviewer #1 minor comments:

*1) The movies are wonderful, but it would be easier to follow them if they were annotated with some labels*.

The revised movies now include labels that identify each of the conformational transitions, as well as each of the structural repeats in the periplasmic and transmembrane domains, and the coupling helicesTM2 and TM8.

*2) In the Introduction the authors refer to the membrane distal portion of the periplasmic domain as the 'TolC-dockign domain'*. *Is this naming convention too suggestive in light of the report of the structure of the pump assembly (see*
[8]
*Nature)?*

The term 'TolC-docking domain' was inspired by the studies of Murakami et al. (Nature, 2002). In addition, Tamura et al. (Biochemistry, 2005) detected cross-linking between these two components of the tripartite system, and thus inferred a direct interaction. This conclusion is at odds with the recent EM studies by Du et al. (Nature, 2014), which indicate no direct contact between TolC and AcrB. At the time of the submission of our article, the results of Du et al. had not been made public yet. A change in nomenclature seems appropriate now. Therefore, we refer to this region of the protein as the ‘funnel domain’, as it features the first section of the exit pore extended by AcrA and TolC.

*3)*
Figure 4
*is a new interpretation of old data*. *The density looks a little poor in the L state: are the side chains disordered?*

In actuality the data shown in Figure 4 had not been reported/discussed earlier. The structural model and 2Fo-Fc refined maps correspond to PDB entry 4DX5, which was indeed the basis for an earlier article (Eicher et al., PNAS, 2012). However, this earlier publication was exclusively focused on novel aspects of drug recognition in the porter domain, and not on the mechanism of the transmembrane domain, or its coupling with the porter domain. Regarding the quality of the data, it is true that the L state appears to be more dynamic than the other two states; for example, the atomic B-factors for the residues shown in Figure 4 are in general higher in the L state (between 48 and 78 A^2^) than in the T and O states (between 31 and 60 A^2^). Nevertheless, the assignment of the side-chain conformations is unambiguous in all cases.

*4) Abstract*, *change to read “...one of the most...”*

This has been changed accordingly.

*Reviewer #2 minor comments*:

*1) What do the authors mean by saying that the ...TM domain had not previously described…*?

Please see our answer to question #3 by Reviewer 1. We changed this sentence to “whose TM domain had not been previously analyzed”.

*2)*
Figure 5*: there is an isolated residue in the cytoplasmic side that changes its protonation probability during the cycle. This should be labeled*.

The residue in question (H505) is labeled in the revised figure.

*3)*
Figure 10
*legend. They refer to figure dashed lines and solid lines in*
Figure 8*, should be*
Figure 9.

This has been changed accordingly.

*4) Regarding the suggested role of R971 in blocking proton leakage*: *is there any indication that R971A expression is toxic to the cell?*

We overproduced the R971A variant in *E. coli* C43(DE3) cells for purification and crystallization purposes. We observed no effect on growth compared to cells producing AcrB wildtype. This variant was produced as well in *E. coli* BW25113ΔacrB carrying pET24acrBHis_R971A (Seeger et al., Biochemistry, 2009), to determine MIC values. Under conditions where no drug was added to the growth medium, these cells grew to OD600 values comparable to those cells expressing wildtype AcrB or other variants. There are a number of ways to interpret these results, but we believe the most plausible explanation is that in the absence of drug, AcrB adopts the symmetric LLL state observed in X-ray structures that lack a bound substrate. The absence of a T protomer in this ‘resting state’ of the trimer would explain the lack of H^+^ leakage.

*5) Are the mutants D407N and D408N fully inactive? Could it be that they are moving only one proton at a time and they can then generate smaller gradients? Is it possible to look at AcrB-dependent passive loading of ethidium in whole cells? Is the assay that follows hydrolysis of cephalosporins sensitive enough for this purpose*?

In our experience and within the limitations of the MIC value determination, D407N and D408N clones exhibit identical susceptibility towards all drugs tested. We routinely use passive loading and active efflux experiments with fluorescent dyes to test the efflux activity of AcrB variants and take either D407N or D408N clones as negative controls. Thus far, we have not observed any activity in these assays. The assay based on hydrolysis of cephalosporins (Lim and Nikaido, AAC, 2010) might be indeed more suitable to detect residual activity, since acrAB knock out strains produce only 1% of the efflux activity compared to wildtype. However, this assay requires a special set-up of the spectrophotometer, currently not available in our lab, to reduce scattering caused by the measurements with whole *E. coli* cells.

*6) The model presented here has some very clear predictions that could be challenged experimentally or supported by existing data. Is there any evidence that supports a stoichiometry of 2H+/substrate? As far as I know AcrB transports positively and negatively charged substrates as well as neutral ones. The driving force for them with this stoichiometry would be very different at different pHs. I am aware of the problem of making proteoliposomes that transport substrate but they may be able to make PLs and follow substrate-induced proton transport*.

In the current hypothesis, the tripartite system sequesters drugs from the periplasm/membrane boundary and expels them over the outer membrane. That is, drugs are not transported from the cytoplasm across the plasma membrane, but only across the outer membrane. For protons, however, AcrB mediates classical transport from the periplasm to the cytoplasm, to harness the electrochemical gradient and energize the conformational changes necessary for active efflux of drugs. Consequently, the charge of the transported drugs is not a relevant factor i.e. the overall process of transport will be electrogenic for every drug molecule transported from the periplasm to the outside, regardless of its charge. The thermodynamic quantity that depends on whether each transport cycle is coupled to one or two H^+^ is the magnitude of the substrate gradient sustained by the action of the AcrAB-TolC efflux pump. Comparing MIC data from cells lacking AcrB with those with active AcrB, it appears that the pump is in some cases able to sustain gradients larger than 1:1000, e.g. for the substrate oxacillin (Kobayashi et al., JBC, 2014). The energetic cost required to sustain such gradients would exceed the energy gain associated with the transport of a single proton across a membrane potential of ∼-180 mV. These observations, in combination with the structural information we obtained for the three different states (L and T with no protons bound to Asp407 and Asp408, vs. two protons bound to these residues in the O state) and the two alternating-access pathways deduced by the MD simulation studies, strongly indicate a H^+^:drug stoichiometry of 2:1. Clearly, direct experimental evidence from reconstituted AcrAB-TolC in proteoliposomes would settle this question, but this remains an experimental challenge, due to the complexity of this two-membrane spanning tripartite system.